# PRISM: PROMPT-REFINED IN-CONTEXT SYSTEM MODELING FOR FINANCIAL RETRIEVAL

## ABSTRACT

With the rapid progress of large language models (LLMs), financial information retrieval has become a critical industrial application. Extracting task-relevant information from lengthy financial filings is essential for both operational and analytical decision-making. We present PRISM, a training-free framework that integrates refined system prompting, in-context learning (ICL), and lightweight multi-agent coordination for document and chunk ranking tasks. Our primary contribution is a systematic empirical study that characterizes when each component provides value; prompt engineering delivers consistent performance with minimal overhead, ICL enhances reasoning for complex queries when applied selectively, and multi-agent systems show potential primarily with larger models and careful architectural design. Extensive ablation studies across FinAgentBench, FiQA-2018, and FinanceBench reveal that simpler configurations often outperform complex multi-agent pipelines, providing practical guidance for practitioners. Our best configuration achieves an NDCG@5 of 0.71818 on FinAgentBench, ranking third while being the only training-free approach in the top three. We provide comprehensive feasibility analyses covering latency, token usage, and cost trade-offs to support deployment decisions. Code is ready for public release.

## 1 INTRODUCTION

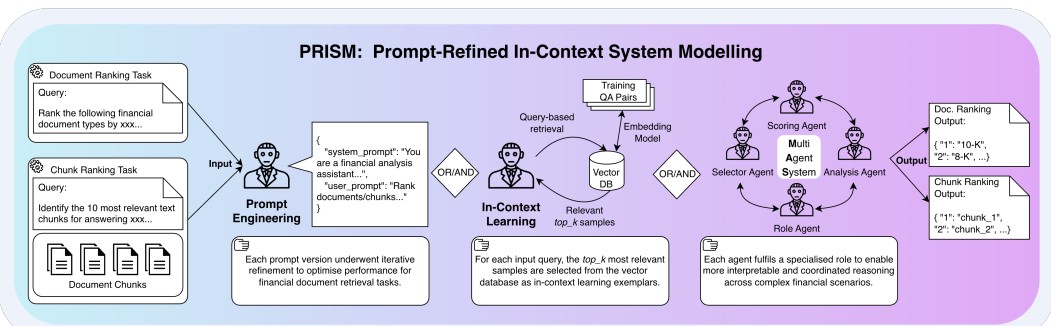

Figure 1: Overview of the PRISM framework. User queries are processed through three independent components: (1) prompt engineering encodes domain priors and reasoning scaffolds, (2) ICL retrieval augments prompts with semantically similar examples from the training set, and (3) optional multi-agent coordination decomposes ranking into specialized agent roles. The system outputs ranked documents and chunks for financial information retrieval.

Large language model (LLM) based Information Retrieval (IR) has emerged as a transformative technology in the financial domain (Zhu et al., 2023). Extracting key information from lengthy financial documents remains labor-intensive and error-prone (Zhao et al., 2024b), driving research toward evidence-based retrieval where systems provide verifiable answers through Retrieval-Augmented Generation (RAG) workflows (Choi et al., 2025a). Despite these advances, challenges persist due to the dense and domain-specific nature of financial texts. FinAgentBench was recently introduced to evaluate two tasks: document ranking to determine the relevance of financial documents, and chunk ranking to identify the most relevant segments within a document (Choi et al., 2025b). Building

on recent advances, we investigate how OpenAI's Generative Pre-trained Transformer (GPT) (Islam et al., 2023) can address these challenges. We propose Prompt-Refined In-Context System Modeling (PRISM), a training-free framework that integrates precise system prompt engineering, in-context learning (ICL) augmentation, and multi-agent system modeling. Our contributions are:

1. We present PRISM, a training-free framework that integrates prompt engineering, in-context learning, and multi-agent coordination for document and chunk ranking tasks.

2. We conduct extensive ablation studies that systematically compare complex multi-agent pipelines against simpler prompt-based strategies, revealing when each component provides value and identifying the conditions under which simpler configurations outperform more sophisticated approaches.

3. We provide comprehensive reproducibility and feasibility analyses, including latency, token usage, and cost metrics, establishing baselines for evaluating accuracy-cost trade-offs in agentic retrieval systems.

## 2 RELATED WORK

**Prompt Engineering.** Recent studies have shown that retrieval performance is highly sensitive to how an LLM is instructed (Zhang et al., 2025b). The initial breakthrough in few-shot learning demonstrated that LLMs can perform tasks using in-context exemplars without gradient updates (Brown et al., 2020). However, traditional few-shot approaches struggled with complex reasoning tasks. This limitation was addressed by Chain-of-Thought (CoT) prompting, which enhances reasoning by guiding LLMs through intermediate natural language reasoning steps (Wei et al., 2022). The Reasoning and Acting (ReAct) framework further improves reliability by allowing LLMs to plan and execute reasoning steps iteratively, reducing hallucination (Yao et al., 2023b). Building on this, the Tree-of-Thoughts (ToT) framework enables LLMs to explore and evaluate multiple reasoning paths to select the next step (Yao et al., 2023a).

**LLMs for Ranking.** LLMs have also demonstrated strong potential as zero-shot rankers in IR, typically categorized into pointwise, pairwise, and listwise approaches (Hou et al., 2024). Listwise methods, such as RankGPT, are often preferred for their balance between effectiveness and efficiency, but they face key challenges including limited input length and sensitivity to document order (Sun et al., 2024). TourRank introduces a multi-stage grouping and tournament strategy to handle large candidate sets and improve robustness to input ordering (Chen et al., 2025). Similarly, Pairwise Ranking Prompting introduces pairwise comparison of output with linear complexity (Qin et al., 2024). Generally, well-instructed LLMs have demonstrated superior ranking performance compared to state-of-the-art supervised systems on major benchmarks such as TREC-DL and BEIR (Sun et al., 2024). However, existing unsupervised methods remain constrained by LLMs' token limits when applied to long and dense financial texts (Zhao et al., 2024a).

**In-Context Learning (ICL).** In-context learning (ICL) enables LLMs to adapt to new tasks using input-output examples in the prompt, without gradient updates. Recent work views this mechanism as implicit Bayesian inference, where the model infers a latent concept that explains the observed examples (Xie et al., 2022). Empirically, ICL accuracy scales with both the number and length of examples but is not strictly tied to correct input-label mappings. Randomly replacing labels only marginally affects performance (Min et al., 2022), suggesting that ICL primarily benefits from contextual cues that define the label space, input distribution, and output format. The emergence of ICL has also been linked to properties of pre-training data such as temporal burstiness and diverse, low-frequency classes (Chan et al., 2022). Meta-ICL aims to refine the model's implicit priors and adaptive strategies by presenting multiple tasks sequentially (Coda-Forno et al., 2023). However, embedding-based ICL retrieval for financial IR with commercial LLMs remains underexplored.

**Agentic Information Retrieval.** Agentic information retrieval represents a new paradigm where LLM agents dynamically manage information access through iterative cycles of observation, reasoning, and action, distinguishing it from traditional IR architectures (Zhang et al., 2025a). It is commonly implemented using multi-agent systems (MAS), which coordinate specialized agents for reflection, planning, and tool use to achieve collective intelligence and outperform single-agent approaches (Singh et al., 2025). LLM agents have also been applied to ranking tasks, demonstrating that retrieval and ranking agents can effectively coexist (Xu et al., 2025). Despite its potential, Agen-

tic IR faces challenges in coordinating complex MAS interactions, reducing the high inference cost of LLMs, and evaluating dynamic agentic behaviors in financial retrieval contexts.

# 3 DATASETS

We evaluate PRISM on three financial datasets that cover complementary task characteristics: ranking granularities (document vs. chunk vs. passage), task types (ranking vs. question answering), and reasoning demands (lexical matching vs. numerical analysis).

**FiQA-2018** (Maia et al., 2018) is a financial opinion QA dataset from the BEIR benchmark (Thakur et al., 2021), containing questions sourced from financial microblogs, news headlines, and reports. The task requires passage reranking to identify relevant answers. NDCG@10 and Recall@100 are used as evaluation metrics, comparing against baselines such as BM25 and cross-encoder rerankers.

**FinanceBench** (Islam et al., 2023) is a question-answering benchmark requiring numerical reasoning over financial documents. Unlike ranking tasks, it tests whether systems can extract correct factual answers from SEC filings. We evaluate under oracle context conditions (relevant passages provided) to isolate the reasoning component from retrieval.

**FinAgentBench** (Choi et al., 2025b) is a large-scale dataset for financial IR comprising 2023–2024 SEC filings from the EDGAR database. It supports two tasks: (i) ranking five document types (10-K, 10-Q, 8-K, DEF 14A, Earnings Transcripts) by relevance to a query, and (ii) ranking text chunks within documents. The dataset includes 4,986 document-ranking and 18,855 chunk-ranking training samples, with 200 validation samples per task.

## 3.1 EXPLORATORY DATA ANALYSIS (EDA) ON FINAGENTBENCH

Table 1: Document distribution across ranks.

| Doc. Type | Rank 1 | Rank 2 | Rank 3 | Rank 4 | Rank 5 | Doc. Type | Rank 1 | Rank 2 | Rank 3 | Rank 4 | Rank 5 |
|---|---|---|---|---|---|---|---|---|---|---|---|
| DEF 14A | 573 | 749 | 1013 | 1212 | **1439** | 10-Q | 554 | **1558** | 1427 | 866 | 581 |
| 10-K | **2402** | 1546 | 507 | 207 | 324 | 8-K | 97 | 407 | 1091 | **1808** | 1583 |
| Earnings | **1360** | 726 | 948 | 893 | 1059 | | | | | | |

Table 2: Top 6 keywords appearing in queries where the respective document type was retrieved at Rank 1. The ratio indicates the frequency of the keyword relative to the entire corpus.

| DEF 14A | | 10-K | | 10-Q | | 8-K | | Earnings | |
|---|---|---|---|---|---|---|---|---|---|
| Keyword | Ratio | Keyword | Ratio | Keyword | Ratio | Keyword | Ratio | Keyword | Ratio |
| dependency | 88.40% | evolved | 40.98% | quarter | 20.00% | compensation | 95.45% | asked | 87.68% |
| concentration | 88.24% | recurring | 40.57% | recurring | 6.29% | burn | 41.74% | questions | 86.56% |
| exist | 87.73% | ratio | 39.40% | evolved | 6.12% | award | 41.74% | metrics | 78.26% |
| risks | 80.51% | time | 39.22% | time | 5.99% | manage | 41.54% | customer | 74.62% |
| market | 60.77% | reporting | 38.19% | ratio | 5.97% | availability | 40.60% | guidance | 52.97% |
| share | 57.12% | period | 36.01% | revenue | 4.02% | share | 27.21% | offered | 52.80% |

**Document Ranking Data Analysis.** We first conduct a systematic evaluation across all document types in FinAgentBench. Table 1 summarizes the distribution of document types across ranking positions. Documents such as 10-K and Earnings Transcripts appear more frequently in higher ranks, while 8-K filings are more evenly distributed across lower ranks, suggesting that document types vary in information density and relevance. Table 2 shows the six most frequent keywords found in Rank 1 documents for each type, after removing stopwords. These differences in vocabulary concentration across document types highlight the need for adaptive retrieval strategies; a detailed keyword analysis is provided in Appendix A.1.1.

**Chunk Ranking Data Analysis.** We conducted a frequency analysis to better understand chunk-level characteristics in the dataset. As shown in Figure 2, relevant chunks are generally longer and more information dense than irrelevant ones. The wider interquartile ranges and higher maximum values, along with the long-tail distribution, indicate substantial variability in chunk length, suggesting that adaptive retrieval methods are needed to handle both typical and unusually long chunks. Moreover, the relatively small proportion of relevant chunks highlights the fine-grained and selective nature of financial retrieval. Overall, our EDA shows that relevant answers are concentrated in

specific document types and in a small minority of information-dense chunks, requiring models that can reason with rich content while managing large variations in chunk length.

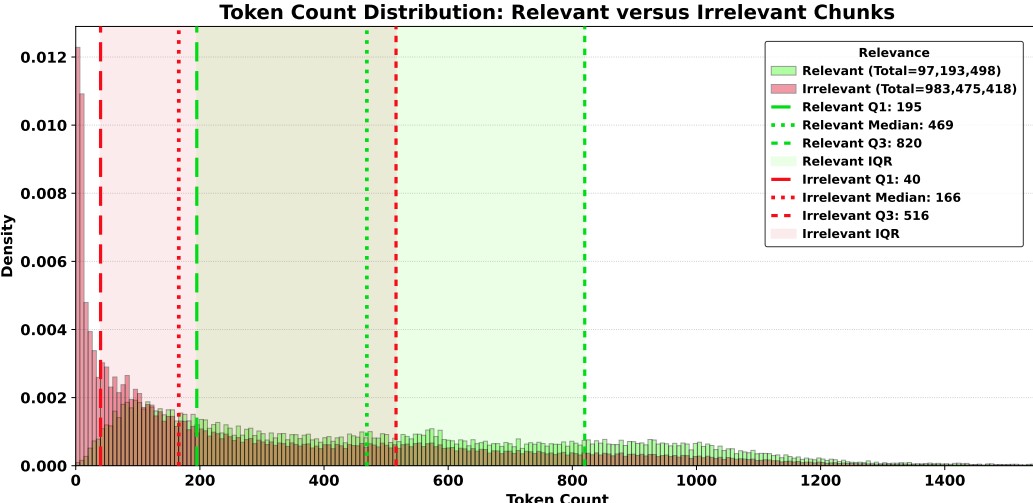

Figure 2: Token count distribution of chunks.

## 4 PRISM

Guided by the EDA findings above, PRISM integrates three components whose interactions are illustrated in Figure 1: (1) prompt engineering that encodes domain priors and reasoning scaffolds, (2) ICL retrieval that dynamically augments prompts with semantically relevant examples, and (3) multi-agent coordination that decomposes ranking into specialized agent roles.

### 4.1 PROMPT ENGINEERING

Prompt engineering in PRISM is an integrated system module that governs how LLMs process ranking tasks. This module encodes domain-specific priors derived from our EDA, such as document-type distributions and keyword patterns, into structured instructions that guide the model's reasoning. We introduce logical scaffolding inspired by ReAct, CoT, and ToT to promote explicit reasoning before producing final answers. Four prompt variants, $P_1$–$P_4$, were systematically designed to assess the impact of incremental modifications and progressively enhance reasoning structure. The $P_1$ variant followed a ReAct-style design with corpus-informed guidance to capture structural regularities in document and chunk distributions. $P_2$ added quantitative domain priors, such as keyword ratios, to complement conceptual cues but showed minimal improvement over $P_1$. $P_3$ incorporated explicit reasoning scaffolds from CoT and ToT to better handle long, information-dense chunks, while $P_4$ streamlined the prompt to be more straightforward by reducing hallucination risk and enforcing stricter output formats for downstream re-ranking. Structural refinements in $P_3$–$P_4$ outperformed content-based augmentations, with $P_4$ achieving the best overall results, confirming that reasoning clarity outweighs informational density. Prompt templates are provided in Appendix A.2.

### 4.2 IN-CONTEXT LEARNING SAMPLE RETRIEVAL

The second module enhances the system with few-shot learning through a simplified RAG pipeline. As our EDA reveals uneven lexical distributions and varying information density across document types and chunks (Tables 1–2), incorporating well-chosen examples that reflect these domain priors gives LLMs a stable reference point, helping them identify relevant documents earlier, reason more consistently, and reduce hallucinations in long-context settings.

**Exemplar Store Construction.** We first construct an offline exemplar store from the training dataset. Each training sample consists of a query paired with documents or chunks and their ground-truth relevance rankings. We embed all query-document/chunk pairs using OpenAI's text-embedding-3 model and index them in a FAISS vector store. This store remains fixed and serves as the source for retrieving few-shot examples.

**Inference-Time Retrieval.** During inference, each new query is embedded using the same model and the top-$k$ most semantically similar exemplars are retrieved from the FAISS store. These exem-

plars, including example queries and their ground-truth rankings, are formatted as few-shot demonstrations and prepended to the system prompt, making each prompt dynamically adapted to the query. Further details on store separation and retrieval dynamics are provided in Appendix A.4.

## 4.3 Multi-Agent System (MAS) Modeling

The third module introduces a structured MAS to decompose reasoning into specialized agent roles, aiming to reduce cognitive load, improve calibration, and mitigate hallucinations through role specialization and graph-level coordination. Despite mixed empirical results, we include MAS to systematically characterize the conditions under which multi-agent coordination provides value (larger models, document-level tasks) versus where it falls short (smaller models, chunk-level tasks with error propagation). This nuanced understanding is a key contribution of our work.

**Architectural Variants.** For chunk ranking, the MAS defines four progressively simplified architectures ($A_1$–$A_4$): $A_1$: Direct scoring ensemble with four parallel agents (CEO, Financial Analyst, Operations Manager, Risk Analyst) that independently evaluate candidate chunks before reaching consensus through score averaging. $A_2$: Three-stage pipeline with Noise Remover $\rightarrow$ Candidate Selector $\rightarrow$ scoring ensemble (Relevance Scorer, Contextual Reasoner, Evidence Extractor, Diversity Agent). This deeper architecture tends to suffer from over-filtering and cascading errors. $A_3$: Two-stage pipeline with Quick Filter $\rightarrow$ three scoring agents (Relevance Scorer, Contextual Reasoner, Evidence Extractor), balancing depth with stability. $A_4$: Minimal configuration with two scoring agents (Financial Analyst, Risk Analyst), reducing interaction complexity and error propagation. Detailed explanations of each variant and its workflow are included in Appendix A.5.

Table 3: Ablation studies of document and chunk ranking configurations across different workflows.

| ID | Document Ranking Configuration | | | Chunk Ranking Configuration | | | Agentic Configuration | | NDCG@5 Score | |
|---|---|---|---|---|---|---|---|---|---|---|
| | Prompt | Model | ICL/Embedding | Prompt | Model | ICL/Embedding | Doc. Agent | Chunk Agent | Public | Private |
| **Non-Agentic Workflow** | | | | | | | | | | |
| 1 | $P_1$ | GPT-4.1 | N/A | $P_1$ | GPT-4.1 | N/A | N/A | | 0.63956 | 0.66666 |
| 2 | $P_1$ | GPT-5-mini | N/A | $P_1$ | GPT-5-mini | N/A | N/A | | 0.65029 | 0.66628 |
| 3 | $P_1$ | GPT-5-mini | N/A | $P_1$ | GPT-4.1 | N/A | N/A | | 0.63834 | 0.66328 |
| 4 | $P_1$ | GPT-4.1 | N/A | $P_1$ | GPT-5-mini | N/A | N/A | | 0.65151 | 0.66966 |
| 5 | $P_1$ | GPT-4.1 | N/A | $P_2$ | GPT-5-mini | N/A | N/A | | 0.63817 | 0.67353 |
| 6 | $P_1$ | GPT-5-mini | N/A | $P_3$ | GPT-5-mini | N/A | N/A | | 0.64711 | 0.67021 |
| 7 | $P_2$ | GPT-5-mini | N/A | $P_1$ | GPT-4.1 | N/A | N/A | | 0.63835 | 0.66088 |
| 8 | $P_2$ | GPT-5-mini | N/A | $P_1$ | GPT-5-mini | N/A | N/A | | 0.63694 | 0.67014 |
| 9 | $P_2$ | GPT-5-mini | N/A | $P_2$ | GPT-5-mini | N/A | N/A | | 0.63695 | 0.66774 |
| 10 | $P_3$ | GPT-5-mini | N/A | $P_1$ | GPT-4.1 | N/A | N/A | | 0.64075 | 0.66673 |
| 11 | $P_3$ | GPT-5-mini | N/A | $P_1$ | GPT-5-mini | N/A | N/A | | 0.6527 | 0.66973 |
| 12 | $P_3$ | GPT-5-mini | N/A | $P_3$ | GPT-5-mini | N/A | N/A | | 0.64297 | 0.69019 |
| 13 | $P_4$ | GPT-5-mini | N/A | $P_4$ | GPT-5-mini | N/A | N/A | | 0.65559 | 0.68895 |
| 14 | $P_4$ | GPT-5-mini | N/A | $P_4$ | GPT-5 | N/A | N/A | | 0.67855 | 0.70575 |
| 15 | $P_4$ | GPT-5 | N/A | $P_4$ | GPT-5 | N/A | N/A | | 0.66236 | 0.70977 |
| 16 | $P_4$ | GPT-5-mini | ICL-5/TE3-S | $P_4$ | GPT-5-mini | N/A | N/A | | 0.66685 | 0.69325 |
| 17 | $P_4$ | GPT-5-mini | ICL-5/TE3-S | $P_4$ | GPT-5-mini | ICL-5/TE3-S | N/A | | 0.64841 | 0.68252 |
| 18 | $P_4$ | GPT-5-mini | ICL-5/TE3-S | $P_4$ | GPT-5 | N/A | N/A | | 0.67373 | 0.71446 |
| **19** | $P_4$ | **GPT-5** | **ICL-5/TE3-S** | $P_4$ | **GPT-5** | **N/A** | **N/A** | | **0.67444** | **0.71818** |
| **Agentic Workflow** | | | | | | | | | | |
| 20 | N/A | GPT-4o-mini | N/A | N/A | GPT-4o-mini | N/A | $A_1$ | | 0.59171 | 0.57276 |
| 21 | $P_1$ | GPT-4o-mini | N/A | $P_1$ | GPT-4o-mini | N/A | $A_1$ | | 0.58714 | 0.58234 |
| 22 | $P_1$ | GPT-4o-mini | N/A | $P_1$ | GPT-4o-mini | N/A | $A_2$ | | 0.52028 | 0.51518 |
| 23 | $P_1$ | GPT-5-mini | N/A | $P_1$ | GPT-5-mini | N/A | $A_2$ | | 0.56317 | 0.55256 |
| 24 | $P_1$ | GPT-4o-mini | N/A | $P_1$ | GPT-4o-mini | N/A | $A_3$ | | 0.53861 | 0.53400 |
| 25 | $P_1$ | GPT-5-mini | N/A | $P_1$ | GPT-5-mini | N/A | $A_3$ | | 0.63515 | 0.66291 |
| 26 | $P_1$ | GPT-4o-mini | N/A | $P_1$ | GPT-4o-mini | N/A | $A_4$ | | 0.52395 | 0.49782 |
| 27 | $P_1$ | GPT-5-mini | N/A | $P_1$ | GPT-5-mini | N/A | $A_4$ | | 0.61226 | 0.63654 |
| 28 | $P_1$ | GPT-5-mini | ICL-5/TE3-S | $P_1$ | GPT-5-mini | ICL-5/TE3-S | $A_4$ | | 0.64721 | 0.62775 |
| 29 | $P_1$ | GPT-5-mini | ICL-10/TE3-S | $P_1$ | GPT-5-mini | ICL-10/TE3-S | $A_4$ | | 0.63089 | 0.62771 |
| 30 | $P_1$ | GPT-5-mini | ICL-10/TE3-L | $P_1$ | GPT-5-mini | ICL-10/TE3-L | $A_4$ | | 0.60038 | 0.62294 |
| 31 | $P_1$ | GPT-5-mini | ICL-15/TE3-L | $P_1$ | GPT-5-mini | ICL-15/TE3-L | $A_4$ | | 0.58428 | 0.59906 |
| 32 | $P_4$ | GPT-5-mini | ICL-10/TE3-S | $P_1$ | GPT-5-mini | ICL-10/TE3-S | $A_4$ | | 0.58328 | 0.55757 |
| **Hybrid Workflow** | | | | | | | | | | |
| 33 | $P_4$ | GPT-5-mini | N/A | $P_4$ | GPT-5 | N/A | $A_3$ | N/A | 0.67825 | 0.70685 |
| 34 | $P_4$ | GPT-5-mini | N/A | $P_4$ | GPT-5 | N/A | $A_4$ | N/A | 0.67234 | 0.69439 |

## 5 Experiments

We evaluated four GPT models (GPT-4o-mini, GPT-4.1, GPT-5-mini, and GPT-5) using OpenAI's text-embedding-3-small (TE3-S) and text-embedding-3-large (TE3-L) for ICL retrieval. Our evaluation spans three datasets selected to cover complementary task characteristics: FinAgentBench (document and chunk ranking with SEC filings), FiQA-2018 (passage reranking for financial QA),

and FinanceBench (factual QA requiring numerical reasoning). While our experiments use OpenAI models exclusively due to infrastructure constraints (see Appendix A), the training-free nature of PRISM allows straightforward adaptation to other LLM providers.

## 5.1 Quantitative Results

**FinAgentBench.** Performance is measured using NDCG@5, with results reported on a validation split of 30% public and 70% private subsets. As ground truth labels for the public and private test sets are not provided, we report only the combined document and chunk ranking scores computed by the Kaggle evaluation system. Following the competition's conclusion, this evaluation system became defunct, preventing further experiments. On the private subset, PRISM ranked 3rd overall while being the only training-free method in the top three. The top two methods employed fine-tuned embedding models (30 epochs and three separate MiniLM models respectively), whereas PRISM achieves competitive performance using only off-the-shelf LLMs with prompt engineering and ICL. Our best observed score across repeated runs of the same configuration (Run 19) is 0.71818, trailing the top method by only 0.007 (0.72497 vs. 0.71818) while leading the remaining methods by approximately 0.02, indicating that training-free approaches can be competitive with fine-tuned systems when properly configured.

Table 4: FinanceBench oracle evaluation results.

Table 5: FiQA evaluation results.

| Prompt | Model | ICL | Correct | Incorrect | Unable to Answer |
|---|---|---|---|---|---|
| Baseline | GPT-5 | N/A | 141 | 9 | 0 |
| $P_1$ | GPT-5 | ICL-9 | 147 | 3 | 0 |
| $P_2$ | GPT-5 | ICL-9 | 142 | 8 | 0 |
| $P_3$ | GPT-5 | ICL-9 | 140 | 10 | 0 |
| $P_4$ | GPT-5 | ICL-9 | 146 | 4 | 0 |

| Prompt | Model | ICL | NDCG@10 | Recall@100 |
|---|---|---|---|---|
| BM25+CE | N/A | N/A | 0.3470 | 0.5390 |
| $P_1$ | GPT-5 | ICL-5 | 0.6104 | 0.8083 |
| $P_2$ | GPT-5 | ICL-5 | 0.6027 | 0.7941 |
| $P_3$ | GPT-5 | ICL-5 | 0.6197 | 0.8206 |
| $P_4$ | GPT-5 | ICL-5 | 0.6097 | 0.8021 |

**FiQA-2018.** As shown in Table 5, PRISM achieves substantial improvements over existing approaches on the test set, with gains of 78.59% in NDCG@10 and 52.24% in Recall@100 compared to the best-performing reranking baseline, BM25+CE. With GPT-5, PRISM consistently achieves NDCG@10 exceeding 0.60 and Recall@100 reaching approximately 0.80 across all prompt variations, indicating robust performance across different configurations. All PRISM configurations outperform the strongest traditional baseline (BM25+CE). Comprehensive model comparisons against all BEIR baselines and feasibility analysis are provided in Appendix Table 9 and Table 10.

**FinanceBench.** PRISM is also evaluated on FinanceBench under oracle context conditions (Table 4), replicating the best-performing baseline from the original paper with a similar model. PRISM achieved 98% accuracy with the $P_1$ prompt configuration, surpassing the baseline. Notably, while prompt variations performed comparably on reranking datasets, the task-specific nature of QA led to $P_1$ slightly outperforming other variants, indicating that optimal prompt formulations are task-dependent. All PRISM configurations exceeded 90% accuracy, demonstrating adaptability across retrieval paradigms. Complete results are provided in Appendix Table 11.

## 5.2 Ablation Studies on FinAgentBench

**System Prompt Engineering.** Based on the results in Table 3, we first analyze the effects of prompt design and model capacity in the non-agentic workflow. Early prompt variants ($P_1$–$P_2$) revealed that concise, direct instructions improved retrieval relevance over overly structured formats. Later versions ($P_3$–$P_4$) emphasized clarity, reasoning focus, and contextual grounding, further stabilizing performance. The $P_4$ prompt, which encouraged explicit reasoning about document relevance while reducing verbosity, produced steady gains and improved the NDCG@5 from 0.64297 (Run 12) to 0.65559 (Run 13) with GPT-5-mini. Increasing model capacity from GPT-4.1 to GPT-5-mini and GPT-5 consistently enhanced performance, showing that larger models follow structured, reasoning-oriented prompts more effectively. Run 15 achieves the best results among these configurations, with systematic reasoning guided by our prompt design to rank documents and chunks accurately.

**In-Context Learning.** The second study examines the impact of ICL on both non-agentic and agentic workflows. In the non-agentic workflow, applying ICL at the document level consistently improved performance, while extending it to both document and chunk levels degraded results due to context overload and fragmented attention. The best configuration applied ICL only at the document stage, achieving the highest score (Run 19, 0.71818), demonstrating that strategically scoped ICL enhances reasoning consistency, whereas excessive context dilutes retrieval focus. In contrast,

the agentic workflow showed mixed results. Increasing the number of shots often reduced performance (Run 28 vs. Runs 29–32), indicating that excessive or poorly selected examples can bias the model's internal policy and impair generalization. A modest setup, such as ICL-5 with GPT-5-mini (Run 28), provided balanced gains and shows that limited examples can guide reasoning without over-constraining it. Overall, ICL is most effective when applied selectively as structured guidance to improve alignment in financial IR tasks.

**Multi-Agent Workflows.** The third study examines the performance dynamics of MAS under different model scales and graph configurations ($A_1$–$A_4$). Each design varies in the number and connectivity of agents handling filtering, ranking, and scoring. Results show that MAS performance is highly sensitive to model size and architectural complexity. Smaller models, such as GPT-4o-mini, struggled in deeper graphs like $A_2$ (Run 22, 0.51518) due to error propagation across agents. In contrast, scaling to GPT-5-mini significantly improved stability and accuracy, with the $A_3$ workflow rising from 0.53400 (Run 24) to 0.66291 (Run 25). This highlights that larger models better sustain coherent multi-agent coordination across complex reasoning chains. Importantly, this model-size dependency is itself a key empirical finding, as it reveals that MAS benefits emerge only when individual agents possess sufficient reasoning capacity to maintain coherent intermediate outputs. For smaller models, the coordination overhead outweighs the benefits of task decomposition. Additional ablations further reveal that hybrid configurations which is agentic at the document level but non-agentic at the chunk level achieved strong results (0.70685, Run 33; 0.69439, Run 34), comparable to the best overall performance. This indicates that document-level agentic reasoning is effective in isolation, whereas extending agentic control to chunk ranking introduces excessive coordination overhead. More findings are reported in Appendix A.11 and Appendix A.

# 6 FEASIBILITY AND REPRODUCIBILITY ANALYSIS

While prior work on FinAgentBench focuses exclusively on ranking accuracy, practical deployment requires understanding computational costs. To our knowledge, we provide the first latency and cost analysis for this benchmark, which also establishes a baseline for evaluating accuracy-latency-cost trade-offs in future systems. Run 19 achieves average latencies of 10.43s for document ranking and 131.13s for chunk ranking, with per-query token costs of $0.0155 and $0.2768 respectively. These metrics meet typical enterprise search latency requirements (sub-minute for document retrieval) while the cost structure (∼$0.30 per complex query) remains viable for high-value financial analysis where accuracy outweighs inference costs. Limitations of PRISM are discussed in Appendix A.11.

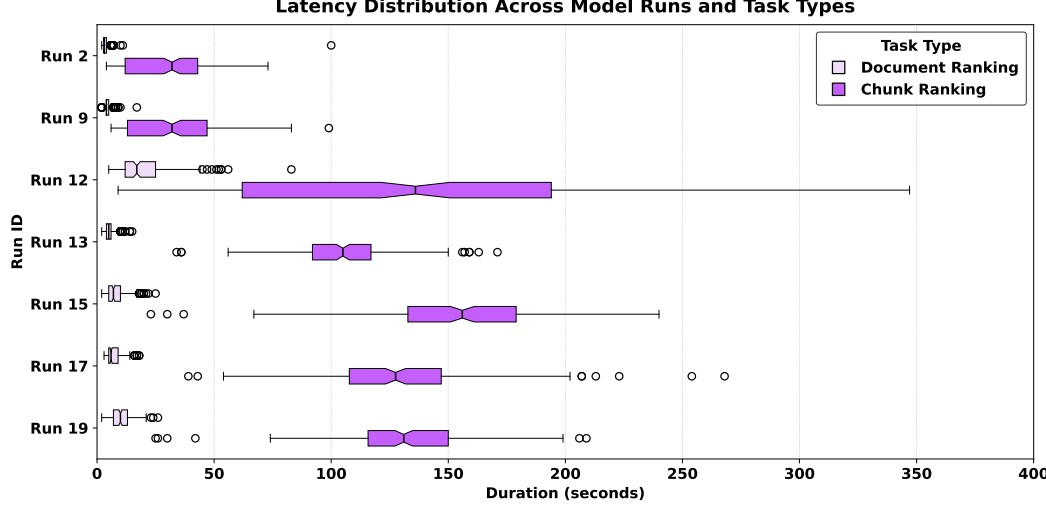

Figure 3: Latency distribution box plots, where each box shows how the latency varies across different runs that are affected by different LLMs and configurations. For chunk ranking tasks, we observe that $P_3$ (Run 12) has the highest median latency for both tasks due to its heavy reasoning prompt design, while $P_4$ (Runs 13–19) achieves a good balance between latency and ranking performance.

**Feasibility Analysis.** Runs of each prompt variant and ICL are selected to evaluate the practical feasibility of PRISM by analyzing latency and token efficiency. Figure 3 shows that document ranking tasks have low median latency (8–10s) with minimal variance, while chunk ranking tasks are slower (130–160s) due to higher reasoning complexity. Runs 15 and 19 strike the best balance between contextual depth and stability. As shown in Figure 4, document-ranking tasks average 2–3K tokens per sample (70% prompt, 30% completion), whereas chunk-ranking tasks use around 100K tokens but remain prompt-dominant (85%). In short, Runs 15 and 19 deliver consistent performance with predictable latency and efficient token usage. Detailed cost analyses are provided in Appendix A.

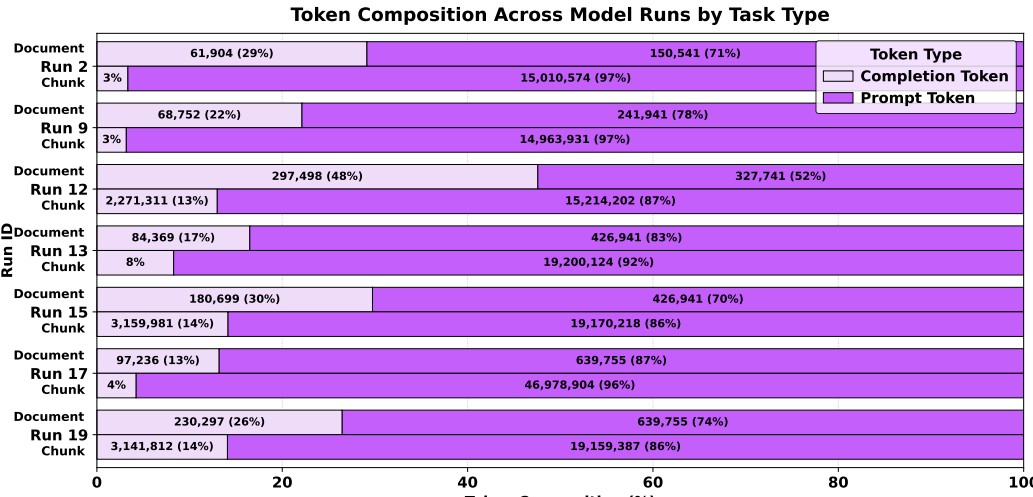

Figure 4: Token distribution bar charts, where each bar details the proportion of prompt and completion token usage across different runs and tasks, highlighting that chunk ranking tasks use more prompt tokens due to longer context and complexity.

**Reproducibility Analysis.** Run 12 serves as the baseline, with Runs 15, 18, and 19 using improved prompts and larger models. As detailed in Tables 6 and 7, all runs exhibit low variability (CV < 1.6%), and Welch's two-sample $t$-tests (Welch, 1938) confirm statistically significant improvements for all comparisons against Run 12 ($p < 0.05$). A statistical discussion is provided in Appendix A.9.

Table 6: Descriptive statistics across multiple runs.

Table 7: Welch's $t$-tests of Run 12.

| Run ID | n | Mean | SD | CV (%) | 95% CI-Low | 95% CI-High |
|---|---|---|---|---|---|---|
| 12 | 5 | 0.68005 | 0.01023 | 1.50489 | 0.66734 | 0.69276 |
| 15 | 4 | 0.70246 | 0.00661 | 0.94078 | 0.69194 | 0.71297 |
| 18 | 4 | 0.70660 | 0.00546 | 0.77319 | 0.69790 | 0.71529 |
| 19 | 9 | 0.71163 | 0.00433 | 0.60861 | 0.70830 | 0.71496 |

| Comparison | $t$-stat | $p$-val | Signif. |
|---|---|---|---|
| 12 vs 15 | -3.969 | 0.0057 | Yes |
| 12 vs 18 | -4.981 | 0.0022 | Yes |
| 12 vs 19 | -6.582 | 0.0014 | Yes |

## 7 CONCLUSION

This work presents PRISM, a training-free framework that integrates system prompting, in-context learning, and multi-agent coordination. Rather than claiming methodological novelty, our primary contribution is a systematic empirical characterization of when each component provides value in financial retrieval. Our extensive ablation studies reveal that simpler non-agentic workflows using the $P_4$ prompt with document-level ICL achieved the strongest overall performance (NDCG@5 of 0.71818), while multi-agent configurations faced coordination overhead and model-size dependencies that limited their effectiveness, challenging common assumptions about the benefits of agentic architectures. Importantly, our study provides practical guidance: prompt engineering offers reliable performance with minimal overhead, ICL enhances reasoning when applied selectively at the document level, and MAS shows potential only with larger models and careful scope limitation. Evaluation across three datasets (FinAgentBench, FiQA-2018, FinanceBench) demonstrates PRISM's generalizability, while our feasibility analysis provides the first latency-cost baselines.

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

## A APPENDIX

In this appendix, we provide additional information and insights that cannot fit into the main paper due to the page limit. We follow the sequence of sections in the main paper for easier understanding.

### A.1 EXPLORATORY DATA ANALYSIS (EDA) ON FINAGENTBENCH

#### A.1.1 DOCUMENT RANKING KEYWORD ANALYSIS

Table 2 reveals distinct keyword concentration patterns across SEC filing types. DEF 14A filings are dominated by high-ratio keywords such as *dependency* (88.40%) and *concentration* (88.24%), while 8-K filings focus more on compensation-related terms (95.45%). 10-Q filings contain notably lower keyword ratios, reflecting greater variation in language use. In contrast, Earnings Transcripts and DEF 14A documents show stronger keyword concentration, suggesting that certain document types have more distinctive language patterns that can help guide retrieval. These patterns informed the design of our domain-specific prompt priors and agent routing logic.

#### A.1.2 CHUNK RANKING DATA ANALYSIS WITH WORD COUNT

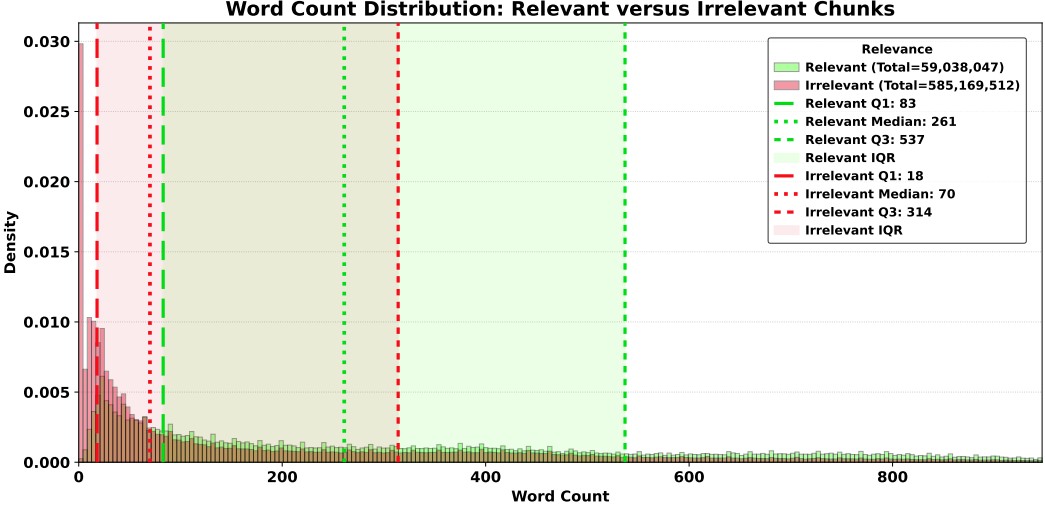

Figure 5: Word count distribution of chunks.

We conducted a frequency analysis on chunk word counts to complement the token count analysis and validate our observations. As shown in Figure 5, the trends closely mirror the token count results in Figure 2 where the relevant chunks consistently contain more words, reflecting their higher informational density and richer semantic content. The similar distribution patterns, including wider interquartile ranges and higher upper extremes, further corroborate the variability observed in chunk lengths. This strong alignment between word and token count analyses reinforces the reliability of our insights. An adaptive retrieval and ranking strategy is needed to identify detailed, content-heavy chunks, with mechanisms to handle large fluctuations in chunk length efficiently.

### A.2 DIFFERENT PROMPT VERSIONS

The non-agentic prompt templates are designed for single LLM workflows, where the model independently performs the complete ranking process using structured system instructions. These prompts emphasize logical consistency, domain alignment, and interpretable reasoning. For document ranking, the progression moves from domain-guided heuristics to combined reasoning frameworks. For chunk ranking, prompts evolve from a pure ReAct framework to integrated multi-strategy approaches. Specifically, the prompt iterations are as follows:

- $P_1$: For document ranking, introduces domain-specific keyword hints and statistical cues to guide relevance assessment. For chunk ranking, employs a ReAct-style framework that interleaves internal reasoning with action steps, requiring the model to think step-by-step before outputting ranked indices.

- $P_2$: For document ranking, enhances $P_1$ with detailed statistical keyword associations (e.g., word frequency ratios per document type) to improve discriminative power. For chunk ranking, combines Chain-of-Thought (CoT), ReAct, and Tree-of-Thought (ToT) strategies, introducing multi-expert simulation where three analysts reason collaboratively and prune incorrect paths.

- $P_3$: Adopts a unified multi-strategy framework for both document and chunk ranking, integrating CoT decomposition, ReAct alternation, and ToT multi-expert simulation. Emphasizes explicit step-by-step reasoning with domain-specific hints for financial concepts.

- $P_4$: Refines the $P_3$ framework with streamlined instructions and consistent terminology across document and chunk ranking tasks, maintaining the combined CoT, ReAct, and ToT reasoning strategies.

The complete prompt templates for document ranking (Table 16) and chunk ranking (Table 17) are provided in Appendix A.12.

## A.3 DYNAMICS OF IN-CONTEXT LEARNING AND PROMPT VERSIONING

ICL was introduced only at the final prompt stage to maintain a clear, experiment-driven workflow. We first focused on stabilizing the prompt design, since it governs the model's reasoning structure and output constraints. Once a stable and effective prompt was established, we introduced ICL as an extension to assess how contextual examples could further enhance model performance. The ICL design evolved progressively, starting with 5 examples to test baseline adaptability, then expanding to 10 and 15 examples to assess the model's ability to leverage larger context windows. This incremental setup allowed us to observe how additional examples influence reasoning stability and retrieval accuracy. By introducing ICL only after the prompt was fully optimized, we ensured that any observed improvements could be attributed directly to the contextual learning mechanism rather than prompt variation, maintaining consistency and preserving interpretability across experiments.

## A.4 IN-CONTEXT LEARNING RETRIEVAL DETAILS

The FAISS store used for ICL exemplar retrieval is separate from the document/chunk corpus that the system ranks; the former provides learning examples while the latter contains the actual candidates to be ranked. This separation ensures that retrieved exemplars serve as reasoning scaffolds without contaminating the ranking candidates. The resulting prompt is dynamic, adapting to the semantic characteristics of each user query rather than relying on a static template. Empirically, we find that ICL performs best when the retrieved samples are semantically close to the target query, as closer exemplars provide more relevant reasoning patterns for the model to follow.

## A.5 MULTI-AGENT SYSTEM ARCHITECTURE DETAILS

This section provides comprehensive implementation details for the Multi-Agent System (MAS) modeling architectures used in PRISM, including the document ranking workflow and four chunk ranking variants ($A_1 - A_4$). All architectures are illustrated in Figure 6.

### A.5.1 AGENT DEFINITIONS

We adopt a state-graph framework implemented using LangGraph, where each agent is an LLM instance with a specialized system prompt defining its role and scoring criteria. Document ranking employs a fixed configuration: a *Question Analyzer* that extracts key entities and intent from the query, followed by five *Document Expert* agents (one per SEC filing type: 10-K, 10-Q, 8-K, DEF 14A, and Earnings Transcripts) that score documents based on domain-specific relevance criteria. For chunk ranking, we define specialized agents including: *Financial Analyst* (evaluates quantitative

metrics and financial data), *Risk Analyst* (assesses operational and regulatory risks), *Evidence Extractor* (identifies concrete supporting facts), *Contextual Reasoner* (evaluates explanatory content), and filtering agents such as *Quick Filter* and *Noise Remover*.

### A.5.2 INTERACTION PROCESS

Agents communicate through a shared state graph where each node represents an agent and edges define information flow. In a typical workflow: (1) input chunks are passed to filtering agents that assign preliminary relevance scores; (2) filtered candidates flow to scoring agents that independently evaluate chunks on a 1–10 scale based on their specialized criteria; (3) scores are aggregated through weighted averaging or voting to produce final rankings. The graph topology determines whether agents operate in parallel (simultaneous scoring) or sequentially (pipeline filtering). All agent prompts are provided in Table 19.

### A.5.3 STATE MANAGEMENT AND INFORMATION FLOW

All MAS architectures maintain a shared state graph implemented through LangGraph's StateGraph abstraction. Information flows through the graph according to predefined edges connecting agent nodes. Parallel agents (e.g., scoring agents in $A_1$) execute simultaneously without inter-agent communication, reducing coordination complexity. Sequential pipelines ($A_2$, $A_3$) enforce strict ordering where downstream agents only receive filtered candidates from upstream stages, creating potential bottlenecks if early filtering is overly aggressive. The state contains the input data, intermediate outputs, and aggregated results.

### A.5.4 DOCUMENT RANKING WORKFLOW

The document ranking workflow (Figure 6a) follows a two-stage architecture designed to handle heterogeneous financial document types with specialized domain expertise. The workflow begins with a *Question Analyzer* agent that processes the user query to extract key entities and classify the query relevance to the corresponding agents' expertise. This structured analysis is then distributed to five parallel *Document Expert* agents, each specialized in a specific SEC filing type:

- 10-K Expert: Evaluates annual reports, focusing on comprehensive financial statements, business operations, and risk factors disclosed in Form 10-K filings.
- 10-Q Expert: Assesses quarterly reports, prioritizing interim financial performance, management discussion and analysis (MD&A), and quarterly updates.
- 8-K Expert: Analyzes current reports for material events, corporate announcements, and time-sensitive disclosures.
- DEF14A Expert: Reviews proxy statements for governance information, executive compensation, and shareholder proposals.
- Earnings Transcript Expert: Evaluates earnings call transcripts for management commentary, forward guidance, and analyst Q&A insights.

Each document expert scores documents on a 0–4 scale based on relevance to the analyzed query intent. Scores are then aggregated through weighted averaging, where weights are determined by document type relevance to the query category provided by the question analyzer. This parallel scoring approach ensures domain-appropriate evaluation while maintaining computational efficiency, as document experts operate simultaneously without sequential dependencies.

### A.5.5 CHUNK RANKING WORKFLOW VARIANTS

The chunk ranking workflows (Figures 6b–6e) represent four architectural variants with progressively simplified designs, each exploring different trade-offs between reasoning depth, error propagation, and computational cost.

$A_1$: Direct Scoring Ensemble (Figure 6b) implements a single-stage parallel scoring approach with four domain-expert agents operating simultaneously:

- *CEO Agent*: Evaluates strategic relevance and high-level business implications.

- *Financial Analyst*: Assesses quantitative metrics, financial ratios, and numerical evidence.
- *Operations Manager*: Focuses on operational details, business processes, and execution aspects.
- *Risk Analyst*: Identifies risk factors, uncertainties, and potential adverse impacts.

Each agent independently scores all candidate chunks on a 1–10 scale according to its specialized criteria. Final rankings are determined through simple averaging of the four scores. This architecture minimizes pipeline depth to reduce error accumulation and analyzes chunks from different perspectives.

$A_2$: Three-Stage Deep Pipeline (Figure 6c) implements the most complex workflow with three sequential stages designed for sequential filtering and specialized evaluation:

1. Noise Removal Stage: A *Noise Remover* agent filters out clearly irrelevant chunks (boilerplate text, legal disclaimers, table fragments) to reduce downstream computational burden.

2. Candidate Selection Stage: A *Candidate Selector* agent performs secondary filtering, retaining only chunks with potential relevance based on keyword matching and semantic proximity.

3. Ensemble Scoring Stage: Four specialized agents score the filtered candidates:
    - *Relevance Scorer*: Direct query-chunk semantic alignment.
    - *Contextual Reasoner*: Evaluates explanatory content and contextual completeness.
    - *Evidence Extractor*: Identifies concrete supporting facts and data points.
    - *Diversity Agent*: Ensures ranking diversity to avoid redundant information.

While this deep architecture aims to progressively refine candidates, our experiments reveal it suffers from over-filtering and cascading errors, where early-stage false negatives cannot be recovered in later stages, leading to degraded recall performance.

$A_3$: Two-Stage Balanced Pipeline (Figure 6d) simplifies $A_2$ by reducing to two stages while maintaining filtering benefits:

1. Quick Filtering Stage: A *Quick Filter* agent performs lightweight relevance assessment, applying looser thresholds than $A_2$'s Noise Remover to preserve recall.

2. Specialized Scoring Stage: Three focused agents evaluate filtered chunks:
    - *Relevance Scorer*: Query-chunk alignment assessment.
    - *Contextual Reasoner*: Contextual adequacy and explanatory quality.
    - *Evidence Extractor*: Concrete evidence identification.

This architecture balances pipeline depth with stability, reducing error propagation risks while maintaining some computational efficiency gains from preliminary filtering.

$A_4$: Minimal Two-Agent System (Figure 6e) represents the simplest viable MAS configuration with direct parallel scoring by two complementary agents:

- *Financial Analyst*: Focuses on quantitative evidence and financial metrics.
- *Risk Analyst*: Evaluates qualitative risk factors and uncertainties.

By eliminating filtering stages and reducing agent count, it minimizes interaction complexity and error propagation pathways. This minimalist design performed comparably to or better than deeper architectures in our experiments, as reduced error propagation outweighs ensemble benefits.

### A.5.6 PROMPT ENGINEERING FOR AGENT ROLES

Each agent receives a specialized system prompt defining its role, evaluation criteria, and output format. For document ranking, expert prompts specify document-type-specific relevance factors. For chunk ranking, scoring agents receive detailed rubrics mapping scores 1–10 to specific evidence characteristics. Filtering agents use binary decision criteria with explicit instructions to prefer recall over precision. All agent prompts are provided in Table 18 and Table 19.

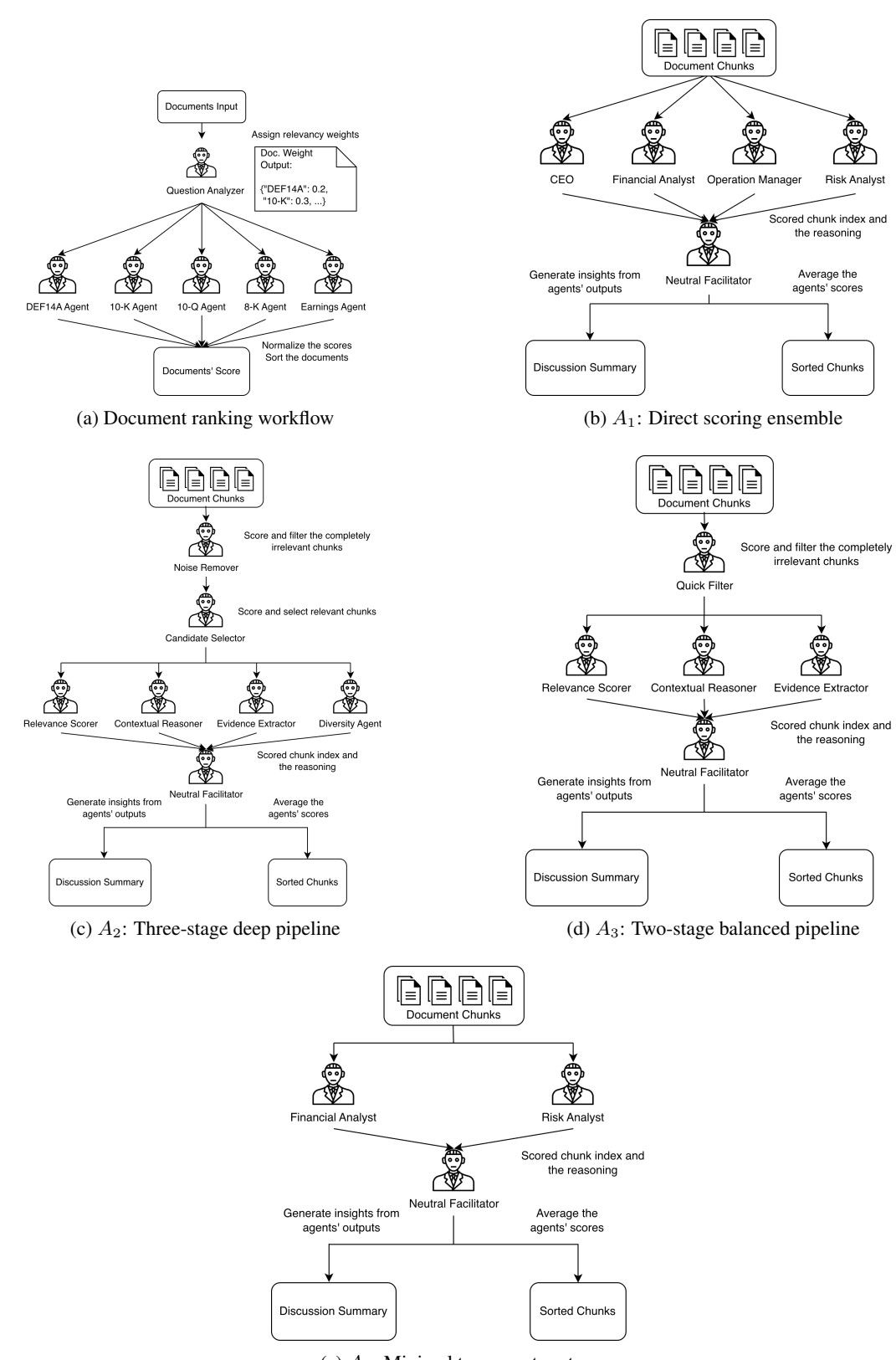

(a) Document ranking workflow

(b) $A_1$: Direct scoring ensemble

(c) $A_2$: Three-stage deep pipeline

(d) $A_3$: Two-stage balanced pipeline

(e) $A_4$: Minimal two-agent system

Figure 6: Multi-Agent System architectures for PRISM. (a) Document ranking workflow. (b)–(e) Chunk ranking variants.

### A.5.7 AGGREGATION AND RANKING LOGIC

Score aggregation varies by architecture. Simple averaging is used in chunk ranking where all agents have equal weight. In document ranking, we implement weighted averaging where weights reflect agent reliability and query-dependent relevance. Final rankings are produced by sorting candidates by aggregated score in descending order, with ties broken by original retrieval rank.

### A.6 IMPLEMENTATION DETAILS

### A.6.1 MODEL PROVIDER SELECTION

All experiments were conducted exclusively using OpenAI models. This decision was driven primarily by practical and infrastructural considerations as most companies in our region operate within the Microsoft Azure ecosystem, which provides direct and optimized access to OpenAI's suite of models. As a startup, we leveraged limited Azure AI credits and native integration to efficiently run large-scale experiments with minimal development and deployment overhead. Beyond operational accessibility, OpenAI's models remain among the most benchmarked closed-source LLMs on financial tasks (Bigeard et al., 2025), offering strong reliability and reproducibility. Therefore, we believe using OpenAI as the sole model provider ensures a stable experimental environment and a credible baseline for future comparisons with alternative LLMs. The exact versions of the four foundation models used in our experiments are provided below:

1. GPT-4o-mini: `gpt-4o-mini-2024-07-18`
2. GPT-4.1: `gpt-4.1-2025-04-14`
3. GPT-5-mini: `gpt-5-mini-2025-08-07`
4. GPT-5: `gpt-5-2025-08-07`

The retrieval pipeline was implemented using a FAISS vector store with two OpenAI embedding backbones: text-embedding-3-small v1 (TE3-S) and text-embedding-3-large (TE3-L). Multi-agent workflows were constructed with LangGraph (v1.0.3), and all models were accessed through the OpenAI Python SDK (v2.3.0).

### A.6.2 LANGGRAPH SELECTION

LangGraph was selected to manage multi-agentic workflows due to its modular design, strong community support, and active maintenance. As a widely adopted framework, it provides a stable and extensible foundation for orchestrating complex agent interactions, allowing developers to define and control the flow of information between agents with precision. Its high degree of customizability enables flexible integration with diverse tools, APIs, and models, making it adaptable to various use cases. Additionally, its emphasis on maintainability and observability supports both rapid prototyping during development and scalable deployment in production. These factors make LangGraph a suitable choice for orchestrating complex MAS financial reasoning pipelines while ensuring long-term sustainability in both research and applied settings.

### A.6.3 PROMPT SELECTION IN AGENTIC WORKFLOW

We fixed the prompt version to $P_1$ for both chunk- and document-ranking tasks in all agentic workflow experiments to ensure controlled and interpretable results. This allowed us to isolate the effects of architectural modifications, ensuring that any performance differences could be attributed to changes in agent coordination rather than confounded by prompt variation. Furthermore, system prompts in a multi-agent setup serve distinct roles across agents, and later prompt versions contain more complex instructions that do not generalize well across heterogeneous agent responsibilities. For example, in Run 32, performance deteriorated when applying the $P_4$ prompt to the document-ranking task, as the increased reasoning complexity introduced confusion among agents. Similarly, attempting to use $P_4$ for chunk-ranking tasks in the hybrid workflow revealed practical constraints as the expanded prompt frequently exceeded token limits due to the large chunk sizes. For these reasons, we maintained a consistent and lightweight prompt configuration while experimenting with different agentic architectures, ensuring comparability, scalability, and clarity in performance attribution.

## A.7 QUANTITATIVE RESULTS

### A.7.1 FINAGENTBENCH

Table 8: FinAgentBench leaderboard results (Private NDCG@5 scores).

| Rank | Team | NDCG@5 Score (Private) | Open Source | Fine-tuning |
|------|------|------------------------|-------------|-------------|
| 1 | memex | 0.72497 | N/A | Yes |
| 2 | Yuzhen Hu | 0.72328 | N/A | Yes |
| 3 | Ours | 0.71181[†] | Yes | No |
| 4 | rlawnsxo | 0.70601 | N/A | N/A |
| 5 | currio | 0.70195 | N/A | N/A |
| 6 | kaggleslasher | 0.69985 | N/A | N/A |
| 7 | Gautam Jajoo | 0.69774 | N/A | N/A |
| 8 | RagTag | 0.69173 | N/A | N/A |
| 9 | ART YOUNG | 0.69162 | N/A | N/A |
| 10 | liangjiang chen | 0.69081 | N/A | N/A |

[†]Score from the submission selected during the competition. Our best observed score across repeated runs of the same configuration (Run 19) is 0.71818, as reported in Table 3.

Quantitative results of the top 10 teams evaluated on the private subset are ranked in Table 8. The top three methods show a small gap between each other and a larger gap compared to the remaining methods. The memex team's solution contains a pool-of-experts of five different LLMs for pre-filtering and 65 agents for chunk retrieval and reranking. Their embedding model is fine-tuned for 30 epochs on the training set of FinAgentBench to improve retrieval precision and better capture question-chunk semantic similarity. The second-ranked Yuzhen Hu team's solution is split into two stages: first retrieving the documents and chunks, then ranking the relevant passages. Three MiniLM models are fine-tuned separately for document retrieval, chunk retrieval, and cross-encoder reranking. In contrast, our PRISM framework is training-free and only leverages off-the-shelf LLMs with well-designed prompts and multi-agent architectures to achieve competitive performance.

**Chunk Ranking Analysis.** While Table 3 reports combined NDCG@5 scores, our detailed analysis reveals that chunk ranking is substantially more challenging than document ranking. Chunk ranking accounts for the majority of latency (131s vs. 10s for documents in Run 19) and token usage (approximately 95K prompt tokens vs. 3K for documents). The performance gap between agentic and non-agentic configurations is most pronounced at the chunk level: MAS architectures consistently underperformed non-agentic approaches for chunk ranking due to error propagation across sequential agent stages. Our best chunk ranking results were achieved with the $P_4$ prompt and GPT-5 without ICL augmentation, suggesting that for fine-grained chunk discrimination, a well-designed single-model approach outperforms distributed multi-agent reasoning. This finding has practical implications: practitioners should prioritize prompt optimization and model selection for chunk ranking, reserving MAS for document-level tasks where coordination overhead is better tolerated.

### A.7.2 FIQA-2018

Table 10 presents comprehensive retrieval performance metrics on FIQA-2018, revealing PRISM's substantial superiority over traditional baselines (Table 9) across all configurations. Most notably, PRISM consistently surpasses the best-performing baseline (BM25+CE) by large margins, with even the lowest-performing PRISM variant achieving better results.

Examining the prompt design progression reveals nuanced performance patterns. $P_1$, employing a straightforward prompting strategy, establishes a strong baseline with GPT-5 achieving 0.6130 NDCG@10 and 0.8045 Recall@100. $P_2$, which integrates key contextual information, demonstrates comparable performance of 0.6095 NDCG@10 and 0.8094 Recall@100 with GPT-5, suggesting that additional context maintains effectiveness while potentially offering enhanced interpretability. $P_3$, utilizing Chain-of-Thought (CoT) reasoning, achieves the highest NDCG@10 of 0.6197, indicating that structured reasoning particularly benefits ranking precision at top positions. Interestingly, $P_4$, which combines ReAct and Tree-of-Thought (ToT) for more explicit multi-step rea-

soning, shows slightly lower performance of 0.6070 NDCG@10, suggesting that excessive prompt complexity may introduce overhead without proportional gains for this task.

The impact of ICL varies significantly by model capability, revealing important insights for practical deployment. ICL proves particularly beneficial for models with lesser reasoning capabilities, as evidenced by GPT-4.1's consistent performance gains across all prompts (e.g., 23.5% improvement in NDCG@10 for $P_1$ configuration). This enhancement occurs because ICL provides concrete demonstrations of the task structure and expected reasoning patterns, effectively compensating for limited inherent reasoning capacity by offering explicit templates to follow. In contrast, GPT-5-mini and GPT-5 exhibit a more stable performance with marginal variations when ICL is applied, indicating their robust capacity to generalize from task instructions alone without requiring extensive demonstration examples unless it is a highly specialized task. This differential ICL sensitivity suggests that smaller or less capable models can be substantially enhanced through careful example selection, while more advanced models already possess sufficient task understanding capabilities.

Overall, all PRISM configurations maintain higher Recall@100 ($\geq 0.80$ for GPT-5) than the best-performing baseline, demonstrating strong retrieval coverage, while improvements in NDCG@10 highlight enhanced precision in ranking highly relevant documents at top positions, a critical capability for practical information retrieval systems.

Table 9: Traditional baseline performance on FIQA-2018.

| Method | Model | NDCG@10 | Recall@100 |
|---|---|---|---|
| BM25 | Baseline | 0.2360 | 0.5390 |
| DeepCT | Sparse | 0.1910 | 0.4890 |
| SPARTA | Sparse | 0.1980 | 0.4460 |
| docT5query | Sparse | 0.2910 | 0.5980 |
| DPR | Dense | 0.1120 | 0.3420 |
| ANCE | Dense | 0.2950 | 0.5810 |
| TAS-B | Dense | 0.3000 | 0.5930 |
| GenQ | Dense | 0.3080 | 0.6180 |
| ColBERT | Late-Interaction | 0.3170 | 0.6030 |
| BM25+CE | Re-ranking | 0.3470 | 0.5390 |

Table 10: PRISM performance on FIQA-2018.

| Prompt | Model | ICL | NDCG@10 | Recall@100 |
|---|---|---|---|---|
| $P_1$ | GPT-4.1 | N/A | 0.4307 | 0.6024 |
| $P_1$ | GPT-5-mini | N/A | 0.5565 | 0.7268 |
| $P_1$ | GPT-5 | N/A | 0.6130 | 0.8045 |
| $P_1$ | GPT-4.1 | ICL-5 | 0.5320 | 0.7247 |
| $P_1$ | GPT-5-mini | ICL-5 | 0.5171 | 0.6412 |
| $P_1$ | GPT-5 | ICL-5 | 0.6104 | 0.8083 |
| $P_2$ | GPT-4.1 | N/A | 0.4871 | 0.6711 |
| $P_2$ | GPT-5-mini | N/A | 0.5521 | 0.7094 |
| $P_2$ | GPT-5 | N/A | 0.6095 | 0.8094 |
| $P_2$ | GPT-4.1 | ICL-5 | 0.4974 | 0.6863 |
| $P_2$ | GPT-5-mini | ICL-5 | 0.4960 | 0.6142 |
| $P_2$ | GPT-5 | ICL-5 | 0.6027 | 0.7941 |
| $P_3$ | GPT-4.1 | N/A | 0.4479 | 0.5948 |
| $P_3$ | GPT-5-mini | N/A | 0.5642 | 0.7374 |
| $P_3$ | GPT-5 | N/A | 0.6094 | 0.8136 |
| $P_3$ | GPT-4.1 | ICL-5 | 0.4702 | 0.6471 |
| $P_3$ | GPT-5-mini | ICL-5 | 0.5182 | 0.6506 |
| $P_3$ | GPT-5 | ICL-5 | 0.6197 | 0.8206 |
| $P_4$ | GPT-4.1 | N/A | 0.4178 | 0.5739 |
| $P_4$ | GPT-5-mini | N/A | 0.5269 | 0.6812 |

Table 10 – *Continued from previous page*

| Prompt | Model | ICL | NDCG@10 | Recall@100 |
|--------|-------|-----|---------|------------|
| $P_4$ | GPT-5 | N/A | 0.6070 | 0.8195 |
| $P_4$ | GPT-4.1 | ICL-5 | 0.4544 | 0.6212 |
| $P_4$ | GPT-5-mini | ICL-5 | 0.4615 | 0.5610 |
| $P_4$ | GPT-5 | ICL-5 | 0.6097 | 0.8021 |

### A.7.3 FINANCEBENCH

To assess PRISM's versatility beyond reranking tasks, we extended our evaluation to FinanceBench, complementing our experiments on the reranking datasets FiQA and FinAgentBench. For a reliable and fair comparison, we replicated the best-performing baseline from the FinanceBench paper, implementing their LLM approach with similar models across multiple evaluation modes. Table 11 presents comprehensive results revealing PRISM's effectiveness in question-answering tasks. Under the Oracle context conditions, PRISM achieved a near-perfect accuracy of 98%, substantially surpassing the baseline performance. More importantly, PRISM demonstrates consistent improvements across all evaluation modes and models. For instance, all PRISM variants ($P_1$–$P_4$) achieve 97%–98% accuracy, representing improvement in an already high-performing regime.

An important observation emerges regarding prompt complexity and task alignment. While $P_4$ showed slightly lower performance in reranking tasks, it maintains competitive performance in QA settings, with GPT-5 Oracle achieving 97.33% accuracy (146/150). This suggests that optimal prompt design is indeed task-dependent, as we observed in the reranking experiments. Simpler prompts ($P_1$) often match or exceed more complex variants, indicating that the structured reasoning scaffolding beneficial for nuanced ranking decisions may be less critical for direct factual retrieval tasks inherent to question-answering.

The limited utility of ICL in FinanceBench can be attributed to task-specific characteristics unlike its substantial benefits in reranking tasks. FinanceBench queries require finding specific answers within individual documents rather than comparing multiple candidates, making similarity-based ICL selection less effective at retrieving truly relevant question-answer exemplars. This results in ICL examples that may misdirect rather than guide the model, explaining why non-ICL configurations often perform comparably or better. Nevertheless, despite the limited dataset size (150 questions) and the already high baseline performance of up to 96% accuracy, PRISM achieves measurable improvements, demonstrating its effectiveness across different task types beyond reranking. While the absolute gains are necessarily constrained in such a high-performing regime approaching the theoretical ceiling of 100%, PRISM's consistent ability to push accuracy toward 98% validates its versatility as a general-purpose framework for LLM-based information retrieval tasks.

Table 11: Comprehensive evaluation results of different prompt versions and models across various evaluation modes on FinanceBench.

| Prompt | Model | Eval Mode | ICL | Results (Out of 150) | | |
|--------|-------|-----------|-----|---------|-----------|------------------|
| | | | | **Correct** | **Incorrect** | **Unable to Answer** |
| **Baselines** | | | | | | |
| Baseline | GPT-4.1 | Single Store | N/A | 132 | 18 | 0 |
| Baseline | GPT-4.1 | Shared Store | N/A | 124 | 26 | 0 |
| Baseline | GPT-4.1 | In Context | N/A | 129 | 21 | 0 |
| Baseline | GPT-4.1 | Oracle | N/A | 133 | 16 | 1 |
| Baseline | GPT-5-mini | Single Store | N/A | 144 | 4 | 2 |
| Baseline | GPT-5-mini | Shared Store | N/A | 143 | 7 | 0 |
| Baseline | GPT-5-mini | In Context | N/A | 142 | 7 | 1 |
| Baseline | GPT-5-mini | Oracle | N/A | 145 | 5 | 0 |
| Baseline | GPT-5 | Single Store | N/A | 141 | 8 | 1 |
| Baseline | GPT-5 | Shared Store | N/A | 143 | 7 | 0 |
| Baseline | GPT-5 | In Context | N/A | 139 | 11 | 0 |

Table 11 – continued from previous page

| Prompt | Model | Eval Mode | ICL | Results (Out of 150) | | |
|--------|-------|-----------|-----|---------|-----------|------------------|
| | | | | **Correct** | **Incorrect** | **Unable to Answer** |
| Baseline | GPT-5 | Oracle | N/A | 141 | 9 | 0 |
| **PRISM** | | | | | | |
| $P_1$ | GPT-4.1 | Single Store | ICL-9 | 124 | 26 | 0 |
| $P_1$ | GPT-4.1 | Shared Store | ICL-9 | 126 | 24 | 0 |
| $P_1$ | GPT-4.1 | In Context | ICL-9 | 124 | 26 | 0 |
| $P_1$ | GPT-4.1 | Oracle | ICL-9 | 131 | 19 | 0 |
| $P_1$ | GPT-5-mini | Single Store | ICL-9 | 139 | 11 | 0 |
| $P_1$ | GPT-5-mini | Shared Store | ICL-9 | 144 | 6 | 0 |
| $P_1$ | GPT-5-mini | In Context | ICL-9 | 140 | 10 | 0 |
| $P_1$ | GPT-5-mini | Oracle | ICL-9 | 143 | 7 | 0 |
| $P_1$ | GPT-5 | Single Store | ICL-9 | 146 | 4 | 0 |
| $P_1$ | GPT-5 | Shared Store | ICL-9 | 145 | 5 | 0 |
| $P_1$ | GPT-5 | In Context | ICL-9 | 136 | 14 | 0 |
| $P_1$ | GPT-5 | Oracle | ICL-9 | 147 | 3 | 0 |
| $P_1$ | GPT-4.1 | Single Store | N/A | 132 | 18 | 0 |
| $P_1$ | GPT-4.1 | Shared Store | N/A | 122 | 27 | 1 |
| $P_1$ | GPT-4.1 | In Context | N/A | 133 | 17 | 0 |
| $P_1$ | GPT-4.1 | Oracle | N/A | 129 | 21 | 0 |
| $P_1$ | GPT-5-mini | Single Store | N/A | 142 | 8 | 0 |
| $P_1$ | GPT-5-mini | Shared Store | N/A | 147 | 3 | 0 |
| $P_1$ | GPT-5-mini | In Context | N/A | 143 | 7 | 0 |
| $P_1$ | GPT-5-mini | Oracle | N/A | 141 | 9 | 0 |
| $P_1$ | GPT-5 | Single Store | N/A | 140 | 10 | 0 |
| $P_1$ | GPT-5 | Shared Store | N/A | 142 | 8 | 0 |
| $P_1$ | GPT-5 | In Context | N/A | 136 | 14 | 0 |
| $P_1$ | GPT-5 | Oracle | N/A | 140 | 10 | 0 |
| $P_2$ | GPT-4.1 | Single Store | ICL-9 | 122 | 28 | 0 |
| $P_2$ | GPT-4.1 | Shared Store | ICL-9 | 114 | 35 | 1 |
| $P_2$ | GPT-4.1 | In Context | ICL-9 | 126 | 24 | 0 |
| $P_2$ | GPT-4.1 | Oracle | ICL-9 | 132 | 18 | 0 |
| $P_2$ | GPT-5-mini | Single Store | ICL-9 | 145 | 5 | 0 |
| $P_2$ | GPT-5-mini | Shared Store | ICL-9 | 133 | 16 | 0 |
| $P_2$ | GPT-5-mini | In Context | ICL-9 | 146 | 4 | 0 |
| $P_2$ | GPT-5-mini | Oracle | ICL-9 | 142 | 7 | 1 |
| $P_2$ | GPT-5 | Single Store | ICL-9 | 139 | 11 | 0 |
| $P_2$ | GPT-5 | Shared Store | ICL-9 | 136 | 12 | 2 |
| $P_2$ | GPT-5 | In Context | ICL-9 | 138 | 12 | 0 |
| $P_2$ | GPT-5 | Oracle | ICL-9 | 142 | 8 | 0 |
| $P_2$ | GPT-4.1 | Single Store | N/A | 130 | 20 | 0 |
| $P_2$ | GPT-4.1 | Shared Store | N/A | 126 | 23 | 1 |
| $P_2$ | GPT-4.1 | In Context | N/A | 126 | 24 | 0 |
| $P_2$ | GPT-4.1 | Oracle | N/A | 140 | 10 | 0 |
| $P_2$ | GPT-5-mini | Single Store | N/A | 145 | 5 | 0 |
| $P_2$ | GPT-5-mini | Shared Store | N/A | 141 | 9 | 0 |
| $P_2$ | GPT-5-mini | In Context | N/A | 140 | 10 | 0 |
| $P_2$ | GPT-5-mini | Oracle | N/A | 141 | 9 | 0 |
| $P_2$ | GPT-5 | Single Store | N/A | 134 | 15 | 1 |
| $P_2$ | GPT-5 | Shared Store | N/A | 144 | 6 | 0 |
| $P_2$ | GPT-5 | In Context | N/A | 140 | 10 | 0 |
| $P_2$ | GPT-5 | Oracle | N/A | 143 | 6 | 1 |
| $P_3$ | GPT-4.1 | Single Store | ICL-9 | 128 | 22 | 0 |

Table 11 – continued from previous page

| Prompt | Model | Eval Mode | ICL | Results (Out of 150) | | |
|--------|-------|-----------|-----|---------|-----------|------------------|
| | | | | Correct | Incorrect | Unable to Answer |
| $P_3$ | GPT-4.1 | Shared Store | ICL-9 | 120 | 30 | 0 |
| $P_3$ | GPT-4.1 | In Context | ICL-9 | 129 | 21 | 0 |
| $P_3$ | GPT-4.1 | Oracle | ICL-9 | 129 | 21 | 0 |
| $P_3$ | GPT-5-mini | Single Store | ICL-9 | 141 | 9 | 0 |
| $P_3$ | GPT-5-mini | Shared Store | ICL-9 | 140 | 10 | 0 |
| $P_3$ | GPT-5-mini | In Context | ICL-9 | 142 | 8 | 0 |
| $P_3$ | GPT-5-mini | Oracle | ICL-9 | 143 | 7 | 0 |
| $P_3$ | GPT-5 | Single Store | ICL-9 | 146 | 3 | 1 |
| $P_3$ | GPT-5 | Shared Store | ICL-9 | 147 | 3 | 0 |
| $P_3$ | GPT-5 | In Context | ICL-9 | 138 | 12 | 0 |
| $P_3$ | GPT-5 | Oracle | ICL-9 | 140 | 10 | 0 |
| $P_3$ | GPT-4.1 | Single Store | N/A | 128 | 22 | 0 |
| $P_3$ | GPT-4.1 | Shared Store | N/A | 121 | 29 | 0 |
| $P_3$ | GPT-4.1 | In Context | N/A | 123 | 27 | 0 |
| $P_3$ | GPT-4.1 | Oracle | N/A | 136 | 14 | 0 |
| $P_3$ | GPT-5-mini | Single Store | N/A | 140 | 10 | 0 |
| $P_3$ | GPT-5-mini | Shared Store | N/A | 140 | 9 | 1 |
| $P_3$ | GPT-5-mini | In Context | N/A | 144 | 6 | 0 |
| $P_3$ | GPT-5-mini | Oracle | N/A | 146 | 0 | 0 |
| $P_3$ | GPT-5 | Single Store | N/A | 133 | 17 | 0 |
| $P_3$ | GPT-5 | Shared Store | N/A | 135 | 13 | 2 |
| $P_3$ | GPT-5 | In Context | N/A | 133 | 17 | 0 |
| $P_3$ | GPT-5 | Oracle | N/A | 139 | 11 | 0 |
| $P_4$ | GPT-4.1 | Single Store | ICL-9 | 120 | 30 | 0 |
| $P_4$ | GPT-4.1 | Shared Store | ICL-9 | 109 | 39 | 2 |
| $P_4$ | GPT-4.1 | In Context | ICL-9 | 125 | 25 | 0 |
| $P_4$ | GPT-4.1 | Oracle | ICL-9 | 128 | 22 | 0 |
| $P_4$ | GPT-5-mini | Single Store | ICL-9 | 138 | 12 | 0 |
| $P_4$ | GPT-5-mini | Shared Store | ICL-9 | 139 | 11 | 0 |
| $P_4$ | GPT-5-mini | In Context | ICL-9 | 143 | 7 | 0 |
| $P_4$ | GPT-5-mini | Oracle | ICL-9 | 140 | 10 | 0 |
| $P_4$ | GPT-5 | Single Store | ICL-9 | 141 | 9 | 0 |
| $P_4$ | GPT-5 | Shared Store | ICL-9 | 135 | 14 | 1 |
| $P_4$ | GPT-5 | In Context | ICL-9 | 137 | 12 | 1 |
| $P_4$ | GPT-5 | Oracle | ICL-9 | 146 | 4 | 0 |
| $P_4$ | GPT-4.1 | Single Store | N/A | 125 | 25 | 0 |
| $P_4$ | GPT-4.1 | Shared Store | N/A | 119 | 31 | 0 |
| $P_4$ | GPT-4.1 | In Context | N/A | 125 | 25 | 0 |
| $P_4$ | GPT-4.1 | Oracle | N/A | 133 | 17 | 0 |
| $P_4$ | GPT-5-mini | Single Store | N/A | 137 | 12 | 1 |
| $P_4$ | GPT-5-mini | Shared Store | N/A | 136 | 12 | 2 |
| $P_4$ | GPT-5-mini | In Context | N/A | 138 | 12 | 0 |
| $P_4$ | GPT-5-mini | Oracle | N/A | 137 | 13 | 0 |
| $P_4$ | GPT-5 | Single Store | N/A | 137 | 13 | 0 |
| $P_4$ | GPT-5 | Shared Store | N/A | 136 | 13 | 1 |
| $P_4$ | GPT-5 | In Context | N/A | 137 | 13 | 0 |
| $P_4$ | GPT-5 | Oracle | N/A | 142 | 8 | 0 |

Table 12: Latency, token usage, and cost statistics for different runs used in feasibility analysis on FinAgentBench.

| ID | Task | Latency Statistics (seconds) | | | | | Token Usage Statistics | | | | Cost Statistics (USD) | | |
|----|------|-----|-------|--------|-------|------|------------------|--------------------|-------------|-----------------|-------------|-----------------|-----------------|
| | | Min | $Q_1$ | Median | $Q_3$ | Max | $\sum_{prompt}$ | $\sum_{completion}$ | $\mu_{prompt}$ | $\mu_{completion}$ | $\sum_{prompt}$ | $\sum_{completion}$ | $\sum_{combined}$ |
| 2 | Document | 2 | 3 | 3 | 4 | 100 | 150,541 | 61,904 | 752.71 | 309.52 | 0.04 | 0.12 | 0.16 |
| 2 | Chunk | 4 | 12 | 32 | 43 | 4190 | 15,010,574 | 519,830 | 75,052.87 | 2,599.15 | 3.75 | 1.04 | 4.79 |
| 2 | Combined | 2 | 3 | 7 | 32 | 4190 | 15,161,115 | 581,734 | 37,902.79 | 1,454.34 | 3.79 | 1.16 | 4.95 |
| 9 | Document | 2 | 4 | 4 | 5 | 17 | 241,941 | 68,752 | 1,209.71 | 343.76 | 0.06 | 0.14 | 0.20 |
| 9 | Chunk | 6 | 13 | 32 | 47 | 99 | 14,963,931 | 490,556 | 74,819.66 | 2,452.78 | 3.74 | 0.98 | 4.72 |
| 9 | Combined | 2 | 4 | 8 | 32 | 99 | 15,205,872 | 559,308 | 38,014.68 | 1,398.27 | 3.80 | 1.12 | 4.92 |
| 12 | Document | 5 | 12 | 17 | 25 | 83 | 327,741 | 297,498 | 1,638.71 | 1,487.49 | 0.08 | 0.59 | 0.68 |
| 12 | Chunk | 9 | 62 | 136 | 194 | 665 | 15,214,202 | 2,271,311 | 76,071.01 | 11,356.56 | 3.80 | 4.54 | 8.35 |
| 12 | Combined | 5 | 17 | 38 | 135 | 665 | 15,541,943 | 2,568,809 | 38,854.86 | 6,422.02 | 3.89 | 5.14 | 9.02 |
| 13 | Document | 2 | 4 | 5 | 6 | 15 | 426,369 | 84,369 | 2,134.71 | 421.85 | 0.11 | 0.17 | 0.28 |
| 13 | Chunk | 34 | 92 | 105 | 117 | 171 | 19,200,124 | 1,734,012 | 96,000.62 | 8,670.06 | 4.80 | 3.47 | 8.27 |
| 13 | Combined | 2 | 5 | 25 | 105 | 171 | 19,627,065 | 1,818,381 | 49,067.66 | 4,545.95 | 4.91 | 3.64 | 8.54 |
| 15 | Document | 2 | 5 | 7 | 10 | 25 | 426,941 | 180,699 | 2,134.71 | 903.50 | 0.53 | 1.81 | 2.34 |
| 15 | Chunk | 23 | 133 | 156 | 179 | 240 | 19,170,218 | 3,159,981 | 95,851.09 | 15,799.91 | 23.96 | 31.60 | 55.56 |
| 15 | Combined | 2 | 7 | 24 | 156 | 240 | 19,597,159 | 3,340,680 | 48,992.90 | 8,351.70 | 24.50 | 33.41 | 57.90 |
| 17 | Document | 3 | 5 | 6 | 9 | 18 | 639,755 | 97,236 | 3,198.78 | 486.18 | 0.16 | 0.19 | 0.35 |
| 17 | Chunk | 39 | 108 | 128 | 147 | 268 | 46,978,904 | 2,079,562 | 234,894.52 | 10,397.81 | 11.74 | 4.16 | 15.90 |
| 17 | Combined | 3 | 6 | 29 | 127 | 268 | 47,618,659 | 2,176,798 | 119,046.65 | 5,442.00 | 11.90 | 4.35 | 16.26 |
| 19 | Document | 2 | 7 | 10 | 13 | 26 | 639,755 | 230,297 | 3,198.78 | 1,151.49 | 0.80 | 2.30 | 3.10 |
| 19 | Chunk | 25 | 116 | 131 | 150 | 209 | 19,159,387 | 3,141,812 | 95,796.94 | 15,709.06 | 23.95 | 31.42 | 55.37 |
| 19 | Combined | 2 | 10 | 26 | 131 | 209 | 19,799,142 | 3,372,109 | 49,497.86 | 8,430.27 | 24.75 | 33.72 | 58.47 |

## A.8 FEASIBILITY ANALYSIS

### A.8.1 COST ANALYSIS

In Table 12, we report the full set of statistics for the runs selected in our feasibility analysis, covering all prompt variants and the ICL-5/TE3-S configuration within the non-agentic workflow. Additionally, we assess the practical feasibility of the proposed approach by analyzing the cost distribution across the selected runs. Document ranking tasks consistently incur low total costs (below $3), reflecting their modest inference requirements. In contrast, chunk ranking tasks dominate the overall expenditure, with total costs ranging from $4.7 to $55, depending on input size and reasoning depth. Combined task runs follow the same pattern, as their total costs are primarily driven by chunk ranking computations. Among all configurations, Runs 15 and 19 exhibit the highest total costs ($57.9 and $58.5 respectively) due to extensive token generation from the larger GPT-5 model, whereas earlier runs such as Runs 2 and 9 remain significantly more economical ( $5) when using smaller models. These results confirm that document-level ranking remains cost-efficient, while chunk reasoning is more expensive as it offers richer contextual understanding and remains the major contributor to overall cost.

Table 13 presents comprehensive computational cost analysis across all prompt variants and models on FiQA-2018 dataset. GPT-5-mini emerges as a compelling choice for production deployment within the PRISM framework. GPT-5-mini achieves performance between GPT-4.1 and GPT-5 (e.g., $P_1$: 0.5565 NDCG@10) while incurring substantially lower operational costs of approximately $21, compared to $142 for GPT-4.1 and $88 for GPT-5 across the entire evaluation. This represents roughly a $4\times$ cost reduction compared to GPT-5 with only an 8–10% performance trade-off, making GPT-5-mini an optimal balance point for production systems where both performance and cost efficiency are critical considerations. Furthermore, GPT-5-mini demonstrates faster processing times ($\sim$34s) compared to both GPT-4.1 ($\sim$78s) and GPT-5 ($\sim$53s), offering additional latency advantages for real-time applications.

An interesting cost-performance paradox emerges when comparing GPT-5 to GPT-4.1. Despite being a more advanced model, GPT-5 consistently delivers superior performance (0.6130 vs. 0.4307 NDCG@10 for $P_1$ without ICL) while maintaining lower operational costs ($88 vs. $142). This counterintuitive result stems from GPT-5's substantially more efficient token generation patterns. The key lies in GPT-5's architectural improvements that enable more concise reasoning: although GPT-5 generates approximately twice as many completion tokens as GPT-4.1, its significantly lower per-token pricing more than compensates for this increase. Additionally, GPT-5's faster processing time compared to GPT-4.1 suggests better computational efficiency at the infrastructure level. These findings challenge the common assumption that newer, more capable models necessarily incur higher costs, demonstrating instead that architectural efficiency gains can simultaneously improve both accuracy and cost-effectiveness. For organizations prioritizing maximum performance with-

Table 13: Comprehensive latency, token usage and cost statistics for different prompt versions and models on FiQA-2018.

| Prompt | Model | ICL | Prompt Tokens (M) | Completion Tokens (K) | Time (s) | Cost (USD) |
|---|---|---|---|---|---|---|
| $P_1$ | GPT-4.1 | N/A | 70.02 | 237.63 | 71.70 | 141.90 |
| $P_1$ | GPT-5-mini | N/A | 82.44 | 293.29 | 38.63 | 21.20 |
| $P_1$ | GPT-5 | N/A | 66.27 | 479.44 | 51.17 | 87.63 |
| $P_1$ | GPT-4.1 | ICL-5 | 73.02 | 249.23 | 62.19 | 148.04 |
| $P_1$ | GPT-5-mini | ICL-5 | 82.58 | 234.98 | 31.20 | 21.11 |
| $P_1$ | GPT-5 | ICL-5 | 82.58 | 474.73 | 56.23 | 107.97 |
| $P_2$ | GPT-4.1 | N/A | 61.60 | 442.55 | 85.43 | 126.74 |
| $P_2$ | GPT-5-mini | N/A | 82.42 | 480.63 | 35.75 | 21.57 |
| $P_2$ | GPT-5 | N/A | 82.42 | 790.47 | 54.24 | 110.93 |
| $P_2$ | GPT-4.1 | ICL-5 | 60.85 | 396.73 | 81.37 | 124.87 |
| $P_2$ | GPT-5-mini | ICL-5 | 82.56 | 389.64 | 29.29 | 21.42 |
| $P_2$ | GPT-5 | ICL-5 | 82.47 | 787.34 | 53.78 | 110.96 |
| $P_3$ | GPT-4.1 | N/A | 40.40 | 344.30 | 82.18 | 83.55 |
| $P_3$ | GPT-5-mini | N/A | 82.43 | 505.36 | 35.09 | 21.62 |
| $P_3$ | GPT-5 | N/A | 82.30 | 606.21 | 49.82 | 108.94 |
| $P_3$ | GPT-4.1 | ICL-5 | 58.99 | 328.15 | 77.96 | 120.60 |
| $P_3$ | GPT-5-mini | ICL-5 | 82.57 | 442.93 | 29.32 | 21.53 |
| $P_3$ | GPT-5 | ICL-5 | 82.57 | 607.09 | 48.14 | 109.28 |
| $P_4$ | GPT-4.1 | N/A | 38.70 | 272.99 | 73.19 | 79.58 |
| $P_4$ | GPT-5-mini | N/A | 82.42 | 341.96 | 34.59 | 21.29 |
| $P_4$ | GPT-5 | N/A | 82.30 | 511.02 | 55.24 | 107.98 |
| $P_4$ | GPT-4.1 | ICL-5 | 43.11 | 283.30 | 86.78 | 88.48 |
| $P_4$ | GPT-5-mini | ICL-5 | 82.56 | 256.27 | 29.02 | 21.15 |
| $P_4$ | GPT-5 | ICL-5 | 82.56 | 516.58 | 52.37 | 108.37 |

out cost constraints, GPT-5 represents the clear optimal choice, delivering state-of-the-art results at lower total cost than the previous generation model.

Table 14: Comprehensive latency, token usage and cost statistics for different prompt versions and models across various evaluation modes on FinanceBench.

| Prompt | Model | Eval Mode | ICL | Prompt Tokens (K) | Completion Tokens (K) | Time (s) | Cost (USD) |
|---|---|---|---|---|---|---|---|
| **Baselines** | | | | | | | |
| Baseline | GPT-4.1 | Single Store | N/A | 4.81 | 27.13 | 494.94 | 0.23 |
| Baseline | GPT-4.1 | Shared Store | N/A | 4.81 | 27.72 | 478.99 | 0.23 |
| Baseline | GPT-4.1 | In Context | N/A | 1260.12 | 40.70 | 2466.77 | 25.53 |
| Baseline | GPT-4.1 | Oracle | N/A | 159.80 | 27.08 | 329.78 | 0.54 |
| Baseline | GPT-5-mini | Single Store | N/A | 4.81 | 18.39 | 1165.15 | 0.04 |
| Baseline | GPT-5-mini | Shared Store | N/A | 4.81 | 18.69 | 1149.36 | 0.039 |
| Baseline | GPT-5-mini | In Context | N/A | 1260.03 | 131.32 | 1505.37 | 3.41 |
| Baseline | GPT-5-mini | Oracle | N/A | 158.90 | 138.23 | 1200.31 | 0.32 |
| Baseline | GPT-5 | Single Store | N/A | 4.81 | 13.33 | 1556.93 | 0.14 |
| Baseline | GPT-5 | Shared Store | N/A | 4.81 | 13.81 | 1674.66 | 0.14 |
| Baseline | GPT-5 | In Context | N/A | 1260.03 | 147.64 | 2066.25 | 17.23 |
| Baseline | GPT-5 | Oracle | N/A | 158.90 | 179.69 | 1433.36 | 2.00 |
| **PRISM** | | | | | | | |
| $P_1$ | GPT-4.1 | Single Store | ICL-9 | 120.91 | 15.40 | 371.14 | 0.37 |
| $P_1$ | GPT-4.1 | Shared Store | ICL-9 | 154.02 | 14.59 | 332.71 | 0.42 |
| $P_1$ | GPT-4.1 | In Context | ICL-9 | 9885.98 | 36.66 | 2609.96 | 20.07 |
| $P_1$ | GPT-4.1 | Oracle | ICL-9 | 377.00 | 26.53 | 285.59 | 0.97 |
| $P_1$ | GPT-5-mini | Single Store | ICL-9 | 105.78 | 15.58 | 1649.79 | 0.06 |

Table 14 – *Continued from previous page*

| Prompt | Model | Eval Mode | ICL | Prompt Tokens (K) | Completion Tokens (K) | Time (s) | Cost (USD) |
|---|---|---|---|---|---|---|---|
| $P_1$ | GPT-5-mini | Shared Store | ICL-9 | 112.44 | 14.01 | 1579.32 | 0.06 |
| $P_1$ | GPT-5-mini | In Context | ICL-9 | 9868.73 | 130.71 | 1368.42 | 2.73 |
| $P_1$ | GPT-5-mini | Oracle | ICL-9 | 352.25 | 100.56 | 1104.98 | 0.29 |
| $P_1$ | GPT-5 | Single Store | ICL-9 | 138.01 | 13.68 | 1836.46 | 0.31 |
| $P_1$ | GPT-5 | Shared Store | ICL-9 | 134.41 | 12.32 | 1834.41 | 0.29 |
| $P_1$ | GPT-5 | In Context | ICL-9 | 9929.93 | 187.48 | 2586.63 | 14.29 |
| $P_1$ | GPT-5 | Oracle | ICL-9 | 341.00 | 166.67 | 1683.26 | 2.09 |
| $P_1$ | GPT-4.1 | Single Store | N/A | 4.81 | 17.99 | 404.56 | 0.15 |
| $P_1$ | GPT-4.1 | Shared Store | N/A | 4.81 | 16.73 | 373.95 | 0.14 |
| $P_1$ | GPT-4.1 | In Context | N/A | 9744.98 | 39.30 | 1584.33 | 19.80 |
| $P_1$ | GPT-4.1 | Oracle | N/A | 192.80 | 26.96 | 281.26 | 0.60 |
| $P_1$ | GPT-5-mini | Single Store | N/A | 4.81 | 17.12 | 1660.00 | 0.04 |
| $P_1$ | GPT-5-mini | Shared Store | N/A | 4.81 | 16.22 | 1574.42 | 0.03 |
| $P_1$ | GPT-5-mini | In Context | N/A | 9744.53 | 147.37 | 1484.62 | 2.73 |
| $P_1$ | GPT-5-mini | Oracle | N/A | 192.35 | 113.48 | 1274.31 | 0.28 |
| $P_1$ | GPT-5 | Single Store | N/A | 4.81 | 14.16 | 1872.66 | 0.15 |
| $P_1$ | GPT-5 | Shared Store | N/A | 4.81 | 13.98 | 1914.46 | 0.15 |
| $P_1$ | GPT-5 | In Context | N/A | 9744.53 | 191.30 | 2651.92 | 14.09 |
| $P_1$ | GPT-5 | Oracle | N/A | 192.35 | 187.49 | 1887.10 | 2.12 |
| $P_2$ | GPT-4.1 | Single Store | ICL-9 | 124.14 | 22.52 | 501.14 | 0.43 |
| $P_2$ | GPT-4.1 | Shared Store | ICL-9 | 104.34 | 18.15 | 351.93 | 0.35 |
| $P_2$ | GPT-4.1 | In Context | ICL-9 | 9859.73 | 41.07 | 1492.38 | 20.05 |
| $P_2$ | GPT-4.1 | Oracle | ICL-9 | 335.20 | 28.47 | 468.11 | 0.90 |
| $P_2$ | GPT-5-mini | Single Store | ICL-9 | 124.14 | 20.94 | 1636.65 | 0.07 |
| $P_2$ | GPT-5-mini | Shared Store | ICL-9 | 127.56 | 19.75 | 1548.20 | 0.07 |
| $P_2$ | GPT-5-mini | In Context | ICL-9 | 9891.23 | 157.62 | 1600.95 | 2.79 |
| $P_2$ | GPT-5-mini | Oracle | ICL-9 | 343.40 | 125.91 | 1496.93 | 0.34 |
| $P_2$ | GPT-5 | Single Store | ICL-9 | 132.06 | 18.57 | 1867.68 | 0.35 |
| $P_2$ | GPT-5 | Shared Store | ICL-9 | 132.96 | 17.82 | 1937.48 | 0.34 |
| $P_2$ | GPT-5 | In Context | ICL-9 | 9912.68 | 186.49 | 1710.18 | 14.26 |
| $P_2$ | GPT-5 | Oracle | ICL-9 | 378.80 | 176.42 | 1251.60 | 2.24 |
| $P_2$ | GPT-4.1 | Single Store | N/A | 4.81 | 21.43 | 433.74 | 0.18 |
| $P_2$ | GPT-4.1 | Shared Store | N/A | 4.81 | 20.33 | 402.21 | 0.17 |
| $P_2$ | GPT-4.1 | In Context | N/A | 9742.13 | 44.08 | 1469.06 | 19.84 |
| $P_2$ | GPT-4.1 | Oracle | N/A | 189.95 | 30.35 | 335.19 | 0.62 |
| $P_2$ | GPT-5-mini | Single Store | N/A | 4.81 | 23.49 | 1678.87 | 0.05 |
| $P_2$ | GPT-5-mini | Shared Store | N/A | 4.81 | 23.33 | 1723.72 | 0.05 |
| $P_2$ | GPT-5-mini | In Context | N/A | 9741.68 | 171.78 | 1622.31 | 2.78 |
| $P_2$ | GPT-5-mini | Oracle | N/A | 189.50 | 143.25 | 1590.80 | 0.33 |
| $P_2$ | GPT-5 | Single Store | N/A | 4.81 | 19.92 | 2076.89 | 0.21 |
| $P_2$ | GPT-5 | Shared Store | N/A | 4.81 | 19.27 | 2047.74 | 0.20 |
| $P_2$ | GPT-5 | In Context | N/A | 9741.68 | 198.25 | 1794.61 | 14.16 |
| $P_2$ | GPT-5 | Oracle | N/A | 189.50 | 191.22 | 1325.28 | 2.15 |
| $P_3$ | GPT-4.1 | Single Store | ICL-9 | 90.31 | 19.62 | 379.90 | 0.34 |
| $P_3$ | GPT-4.1 | Shared Store | ICL-9 | 128.82 | 23.29 | 430.37 | 0.44 |
| $P_3$ | GPT-4.1 | In Context | ICL-9 | 9908.78 | 55.20 | 1596.56 | 20.26 |
| $P_3$ | GPT-4.1 | Oracle | ICL-9 | 337.70 | 38.70 | 478.79 | 0.99 |
| $P_3$ | GPT-5-mini | Single Store | ICL-9 | 128.82 | 22.09 | 1444.10 | 0.08 |
| $P_3$ | GPT-5-mini | Shared Store | ICL-9 | 115.14 | 21.04 | 1457.49 | 0.07 |
| $P_3$ | GPT-5-mini | In Context | ICL-9 | 9896.18 | 184.37 | 1852.82 | 2.84 |
| $P_3$ | GPT-5-mini | Oracle | ICL-9 | 330.35 | 134.41 | 1502.07 | 0.35 |
| $P_3$ | GPT-5 | Single Store | ICL-9 | 130.81 | 18.69 | 2000.21 | 0.35 |
| $P_3$ | GPT-5 | Shared Store | ICL-9 | 139.81 | 18.96 | 2061.80 | 0.36 |
| $P_3$ | GPT-5 | In Context | ICL-9 | 9882.83 | 200.39 | 1768.46 | 14.36 |
| $P_3$ | GPT-5 | Oracle | ICL-9 | 354.50 | 187.28 | 1244.50 | 2.32 |
| $P_3$ | GPT-4.1 | Single Store | N/A | 4.81 | 24.26 | 466.67 | 0.20 |
| $P_3$ | GPT-4.1 | Shared Store | N/A | 4.81 | 22.01 | 423.23 | 0.19 |

Table 14 – *Continued from previous page*

| Prompt | Model | Eval Mode | ICL | Prompt Tokens (K) | Completion Tokens (K) | Time (s) | Cost (USD) |
|--------|-------|-----------|-----|-------------------|-----------------------|----------|------------|
| $P_3$ | GPT-4.1 | In Context | N/A | 9744.83 | 59.43 | 2204.91 | 19.97 |
| $P_3$ | GPT-4.1 | Oracle | N/A | 192.65 | 40.32 | 491.83 | 0.71 |
| $P_3$ | GPT-5-mini | Single Store | N/A | 4.81 | 26.19 | 1777.55 | 0.05 |
| $P_3$ | GPT-5-mini | Shared Store | N/A | 4.81 | 24.58 | 1448.53 | 0.05 |
| $P_3$ | GPT-5-mini | In Context | N/A | 9744.38 | 195.59 | 1901.49 | 2.83 |
| $P_3$ | GPT-5-mini | Oracle | N/A | 192.20 | 162.52 | 1675.36 | 0.37 |
| $P_3$ | GPT-5 | Single Store | N/A | 4.81 | 21.22 | 2072.32 | 0.22 |
| $P_3$ | GPT-5 | Shared Store | N/A | 4.81 | 20.98 | 2210.27 | 0.22 |
| $P_3$ | GPT-5 | In Context | N/A | 9744.38 | 213.21 | 1876.14 | 14.31 |
| $P_3$ | GPT-5 | Oracle | N/A | 192.20 | 222.01 | 1449.19 | 2.46 |
| $P_4$ | GPT-4.1 | Single Store | ICL-9 | 111.54 | 27.35 | 403.09 | 0.44 |
| $P_4$ | GPT-4.1 | Shared Store | ICL-9 | 115.51 | 28.06 | 390.28 | 0.46 |
| $P_4$ | GPT-4.1 | In Context | ICL-9 | 9908.48 | 79.75 | 1720.44 | 20.45 |
| $P_4$ | GPT-4.1 | Oracle | ICL-9 | 333.65 | 50.99 | 535.05 | 1.08 |
| $P_4$ | GPT-5-mini | Single Store | ICL-9 | 127.21 | 44.54 | 1950.23 | 0.12 |
| $P_4$ | GPT-5-mini | Shared Store | ICL-9 | 124.86 | 43.19 | 2001.19 | 0.12 |
| $P_4$ | GPT-5-mini | In Context | ICL-9 | 9907.43 | 240.14 | 2291.46 | 2.96 |
| $P_4$ | GPT-5-mini | Oracle | ICL-9 | 319.55 | 176.88 | 1747.33 | 0.43 |
| $P_4$ | GPT-5 | Single Store | ICL-9 | 118.21 | 24.77 | 2019.59 | 0.40 |
| $P_4$ | GPT-5 | Shared Store | ICL-9 | 136.02 | 23.82 | 2165.59 | 0.41 |
| $P_4$ | GPT-5 | In Context | ICL-9 | 9905.78 | 231.55 | 2216.82 | 14.70 |
| $P_4$ | GPT-5 | Oracle | ICL-9 | 327.20 | 234.87 | 1777.46 | 2.76 |
| $P_4$ | GPT-4.1 | Single Store | N/A | 4.81 | 28.82 | 521.27 | 0.24 |
| $P_4$ | GPT-4.1 | Shared Store | N/A | 4.81 | 27.21 | 484.88 | 0.23 |
| $P_4$ | GPT-4.1 | In Context | N/A | 9746.48 | 82.14 | 1714.16 | 20.15 |
| $P_4$ | GPT-4.1 | Oracle | N/A | 194.30 | 51.59 | 522.74 | 0.80 |
| $P_4$ | GPT-5-mini | Single Store | N/A | 4.81 | 42.44 | 2156.44 | 0.09 |
| $P_4$ | GPT-5-mini | Shared Store | N/A | 4.81 | 47.24 | 2128.26 | 0.10 |
| $P_4$ | GPT-5-mini | In Context | N/A | 9746.03 | 275.14 | 2597.60 | 2.99 |
| $P_4$ | GPT-5-mini | Oracle | N/A | 193.85 | 194.71 | 2007.18 | 0.44 |
| $P_4$ | GPT-5 | Single Store | N/A | 4.81 | 28.72 | 2761.10 | 0.29 |
| $P_4$ | GPT-5 | Shared Store | N/A | 4.81 | 31.62 | 2893.80 | 0.32 |
| $P_4$ | GPT-5 | In Context | N/A | 9746.03 | 246.76 | 2276.21 | 14.65 |
| $P_4$ | GPT-5 | Oracle | N/A | 193.85 | 263.13 | 2043.63 | 2.87 |

Table 14 presents comprehensive computational cost analysis across all prompt variants, models, and evaluation modes on FinanceBench dataset. Single Store and Shared Store modes consistently incur minimal costs that are below \$0.50 for all models, reflecting their limited context requirements with prompt tokens. In contrast, In Context mode dominates the overall expenditure, with costs ranging from \$2.73 to \$25.53 depending on the model, primarily driven by the extensive prompt token requirements needed to embed full document context within the prompt. Oracle mode occupies a middle ground with costs between \$0.28 and \$2.87, as it requires moderate context to provide relevant passages without full document inclusion.

Model selection significantly impacts cost-performance trade-offs. GPT-5-mini emerges as the most cost-efficient option across all evaluation modes, with Oracle mode costing merely \$0.28–\$0.44 compared to \$0.60–\$2.87 for GPT-5 and \$0.80–\$1.08 for GPT-4.1 under similar conditions. Despite this dramatic cost advantage, GPT-5-mini maintains superior performance, achieving 98% accuracy in several configurations. GPT-5, while incurring higher costs, consistently outperforms other models in most configurations. GPT-4.1 exhibits the least favorable cost-performance profile, with comparable costs to GPT-5 but consistently lower accuracy.

Prompt complexity introduces additional computational overhead with varying returns. The straightforward prompt $P_1$ achieves optimal cost-efficiency, maintaining low completion token generation while delivering top-tier performance. More complex prompts like $P_4$ generate substantially more completion tokens, increasing costs without proportional performance gains in most evaluation modes. The impact of ICL is particularly pronounced in In Context mode, where it reduces costs slightly for GPT-5 by providing better task guidance that reduces completion token generation.

Processing time analysis reveals GPT-4.1 as the fastest model in Single Store and Shared Store modes but shows significant variability in other modes. GPT-5 and GPT-5-mini show consistent processing times across different modes, with GPT-5-mini being slightly faster. These results confirm that for production deployment on QA tasks, In Context mode's substantial cost premium makes it impractical for large-scale applications despite marginal performance benefits. GPT-5-mini with simpler prompts offers the optimal balance of accuracy, cost, and latency.

## A.9 REPRODUCIBILITY ANALYSIS

As reported in Tables 6 and 7, all runs exhibit low variability with standard deviations ($s < 0.011$) and coefficients of variation ($\text{CV} < 1.6\%$), indicating stable and reproducible outcomes. The narrow 95% confidence intervals (CI) confirm the statistical reliability of the mean performance estimates. Run 12 recorded the lowest mean score (0.68005), while Runs 15–19 achieved consistently higher averages, with non-overlapping CIs showing statistically significant gains following configuration updates. Welch's two-sample $t$-tests further support these improvements: Run 12 vs. Run 15 ($t = -3.969$, $p = 0.0057$), Run 12 vs. Run 18 ($t = -4.981$, $p = 0.0022$), and Run 12 vs. Run 19 ($t = -6.582$, $p = 0.0014$). The progressively decreasing $p$-values and increasing $t$-statistics indicate that each subsequent configuration yields more pronounced improvements. In general, the low variance and narrow confidence bounds demonstrate that the updated configurations produce statistically significant and reproducible results.

### A.9.1 DESCRIPTIVE STATISTICS

The following statistical metrics were computed for each configuration based on $n$ independent runs:

$$\bar{x} = \frac{1}{n} \sum_{i=1}^{n} x_i \tag{1}$$

where $x_i$ denotes the performance metric from the $i^{\text{th}}$ run. The mean $\bar{x}$ represents the average model performance.

$$\text{SD} = \sqrt{\frac{1}{n-1} \sum_{i=1}^{n} (x_i - \bar{x})^2} \tag{2}$$

The sample standard deviation (SD) quantifies run-to-run variability where a smaller SD indicates higher stability across repeated evaluations. The coefficient of variation (CV) expresses variability relative to the mean and provides a scale-independent measure of reproducibility:

$$\text{CV} = \frac{s}{\bar{x}} \times 100\% \tag{3}$$

A CV below 2% is generally regarded as evidence of excellent consistency across runs. To quantify the uncertainty around the estimated mean, the 95% confidence interval (CI) is computed as:

$$\text{CI}_{95\%} = \bar{x} \pm t_{1-\frac{\alpha}{2}, \, n-1} \times \frac{s}{\sqrt{n}}, \quad \text{where } \alpha = 0.05 \tag{4}$$

Specifically, $t_{1-\frac{\alpha}{2}, \, n-1}$ denotes the critical value of the Student's $t$-distribution with $(n-1)$ degrees of freedom corresponding to a 95% confidence level. The value of $t_{1-\frac{\alpha}{2}, \, n-1}$ can be obtained from a standard $t$-distribution table or computed numerically as:

$$t_{1-\frac{\alpha}{2}, \, n-1} = \text{quantile}_t \left(1 - \tfrac{\alpha}{2}, \text{df} = n-1\right). \tag{5}$$

All configurations exhibit low standard deviations ($s < 0.011$) and coefficients of variation below 1.6%, indicating highly stable and reproducible results across runs. The 95% confidence intervals are narrow, confirming that the estimated means are statistically reliable. Run 12 achieves the lowest mean performance ($\bar{x} = 0.68005$), while Runs 15–19 yield consistently higher means (0.70246–0.71163). The non-overlapping confidence intervals between Run 12 and the later runs demonstrate that the configuration changes introduced after Run 12 resulted in a statistically meaningful performance gain. Subsequent runs (15–19) show only minor numerical variation, suggesting that performance has stabilized and that the system's behavior is reproducible.

Table 15: Runs that are used for statistical significance experiment.

| ID | Document Ranking Configuration | | | Chunk Ranking Configuration | | | Agentic Configuration | | NDCG@5 Score | |
|---|---|---|---|---|---|---|---|---|---|---|
| | Prompt | Model | ICL/Embedding | Prompt | Model | ICL/Embedding | Doc. Agent | Chunk Agent | Public | Private |
| 12 | $P_3$ | GPT-5-mini | N/A | $P_3$ | GPT-5-mini | N/A | N/A | | 0.64144 | 0.66537 |
| 12 | $P_3$ | GPT-5-mini | N/A | $P_3$ | GPT-5-mini | N/A | N/A | | 0.64952 | 0.67366 |
| 12 | $P_3$ | GPT-5-mini | N/A | $P_3$ | GPT-5-mini | N/A | N/A | | 0.65520 | 0.68523 |
| 12 | $P_3$ | GPT-5-mini | N/A | $P_3$ | GPT-5-mini | N/A | N/A | | 0.64193 | 0.68579 |
| 12 | $P_3$ | GPT-5-mini | N/A | $P_3$ | GPT-5-mini | N/A | N/A | | 0.64297 | 0.69019 |
| 15 | $P_4$ | GPT-5 | N/A | $P_4$ | GPT-5 | N/A | N/A | | 0.67047 | 0.69640 |
| 15 | $P_4$ | GPT-5 | N/A | $P_4$ | GPT-5 | N/A | N/A | | 0.66720 | 0.69735 |
| 15 | $P_4$ | GPT-5 | N/A | $P_4$ | GPT-5 | N/A | N/A | | 0.66579 | 0.70630 |
| 15 | $P_4$ | GPT-5 | N/A | $P_4$ | GPT-5 | N/A | N/A | | 0.66236 | 0.70977 |
| 18 | $P_4$ | GPT-5-mini | ICL-5/TE3-S | $P_4$ | GPT-5 | N/A | N/A | | 0.68209 | 0.70187 |
| 18 | $P_4$ | GPT-5-mini | ICL-5/TE3-S | $P_4$ | GPT-5 | N/A | N/A | | 0.67842 | 0.70455 |
| 18 | $P_4$ | GPT-5-mini | ICL-5/TE3-S | $P_4$ | GPT-5 | N/A | N/A | | 0.67514 | 0.70551 |
| 18 | $P_4$ | GPT-5-mini | ICL-5/TE3-S | $P_4$ | GPT-5 | N/A | N/A | | 0.67373 | 0.71446 |
| 19 | $P_4$ | GPT-5 | ICL-5/TE3-S | $P_4$ | GPT-5 | N/A | N/A | | 0.68101 | 0.70491 |
| 19 | $P_4$ | GPT-5 | ICL-5/TE3-S | $P_4$ | GPT-5 | N/A | N/A | | 0.67913 | 0.70827 |
| 19 | $P_4$ | GPT-5 | ICL-5/TE3-S | $P_4$ | GPT-5 | N/A | N/A | | 0.68006 | 0.70865 |
| 19 | $P_4$ | GPT-5 | ICL-5/TE3-S | $P_4$ | GPT-5 | N/A | N/A | | 0.67585 | 0.70923 |
| 19 | $P_4$ | GPT-5 | ICL-5/TE3-S | $P_4$ | GPT-5 | N/A | N/A | | 0.68172 | 0.71181 |
| 19 | $P_4$ | GPT-5 | ICL-5/TE3-S | $P_4$ | GPT-5 | N/A | N/A | | 0.67845 | 0.71277 |
| 19 | $P_4$ | GPT-5 | ICL-5/TE3-S | $P_4$ | GPT-5 | N/A | N/A | | 0.66495 | 0.71375 |
| 19 | $P_4$ | GPT-5 | ICL-5/TE3-S | $P_4$ | GPT-5 | N/A | N/A | | 0.66680 | 0.71713 |
| 19 | $P_4$ | GPT-5 | ICL-5/TE3-S | $P_4$ | GPT-5 | N/A | N/A | | 0.67444 | 0.71818 |

### A.9.2 STATISTICAL SIGNIFICANCE

In order to verify that improvements of PRISM are statistically significant, Welch's two-sample $t$-tests were conducted between Run 12 (baseline configuration) and each subsequent run. The test statistic is defined as:

$$t = \frac{\bar{x}_1 - \bar{x}_2}{\sqrt{\frac{s_1^2}{n_1} + \frac{s_2^2}{n_2}}}, \tag{6}$$

where $\bar{x}_1, s_1, n_1$ and $\bar{x}_2, s_2, n_2$ denote the mean, standard deviation, and sample size of each group. The degrees of freedom are approximated using the Welch-Satterthwaite equation:

$$\nu = \frac{\left(\frac{s_1^2}{n_1} + \frac{s_2^2}{n_2}\right)^2}{\frac{(s_1^2/n_1)^2}{n_1-1} + \frac{(s_2^2/n_2)^2}{n_2-1}}. \tag{7}$$

The two-tailed $p$-value is obtained from the cumulative $t$-distribution:

$$p = 2 \times \mathrm{CDF}_t(-|t|, \nu). \tag{8}$$

A comparison is deemed statistically significant when $p < \alpha$, where $\alpha = 0.05$ corresponds to a 5% significance level with a 5% probability of incorrectly rejecting the null hypothesis.

### A.9.3 INTERPRETATION OF STATISTICAL SIGNIFICANCE

Table 15 summarizes the configurations and corresponding NDCG@5 scores used in the statistical significance experiments. As shown in Table 7, all pairwise comparisons involving Run 12 yield $p$-values below the 0.05 threshold, confirming that every configuration after Run 12 achieves a statistically significant performance improvement over the baseline. Table 6 shows consistently low variability (CV < 1.6%) and narrow confidence intervals across these later runs, demonstrating that the model's performance is both stable and reproducible. These results confirm that the updated configurations produce reproducible and statistically significant improvements in model performance, with minimal run-to-run variance.

### A.10 PRODUCTION READINESS ANALYSIS

This section evaluates the production readiness of the PRISM components across prompt engineering, ICL retrieval, and multi-agent system modeling. While our best-performing configuration relies primarily on prompt engineering and selective ICL, we include all three components to provide a comprehensive empirical characterization of the design space. This approach enables practitioners to make informed decisions about which components to deploy based on their specific constraints and requirements.

### A.10.1 PROMPT ENGINEERING

Prompt engineering is the most mature and deployment-ready component. It delivers stable, low-variance performance with minimal latency and token overhead, making it suitable for production environments that require predictable cost and operational reliability. From an operational standpoint, prompt-based tuning requires no architectural changes and can be easily adapted as new foundation models become available, enabling seamless integration into existing pipelines. Its low latency and minimal token overhead further strengthen its feasibility for production settings.

### A.10.2 IN-CONTEXT LEARNING SAMPLE RETRIEVAL

ICL improves reasoning consistency and retrieval quality but introduces additional computational cost. In production settings, ICL could be selectively activated for complex or high-stakes financial queries where improved contextual understanding outweighs efficiency concerns. When combined with a strong prompt design, ICL can be deployed in controlled configurations, potentially using adaptive strategies that vary the number of retrieved samples based on query complexity or token budget constraints.

### A.10.3 NON-AGENTIC WORKFLOW

This workflow represents the most production-ready configuration within PRISM, leveraging prompt engineering and selective ICL retrieval in a single-LLM architecture. The workflow's production readiness stems from three key advantages. First, it inherits prompt engineering's operational maturity that requires no architectural changes, delivering predictable latency, and adapting seamlessly to new foundation models. Second, it incorporates ICL retrieval selectively, allowing organizations to activate exemplar-based guidance only for complex queries where the performance gains justify additional token costs, maintaining cost efficiency for routine tasks. Third, the single-agent design ensures more deterministic behavior with minimal variance, critical for enterprise deployments requiring consistent service-level agreements.

Our four prompt variants ($P_1$–$P_4$) demonstrate progressive sophistication while maintaining production viability. The modular prompt design enables flexible production deployment strategies where organizations can start with simpler prompts ($P_1$) for cost-sensitive applications, adopt intermediate complexity ($P_2$, $P_3$) for balanced performance-cost trade-offs, or deploy advanced reasoning ($P_4$) for high-stakes queries. Combined with adaptive ICL activation varying exemplar count based on query complexity or token budgets, the non-agentic workflow provides a scalable, operationally reliable foundation suitable for immediate enterprise integration.

### A.10.4 AGENTIC WORKFLOW

The agentic workflow extends the non-agentic prompt scaffolding to a multi-role architecture, where specialized agents inherit the underlying ReAct reasoning framework while contributing distinct domain-specific perspectives. Each agent scores chunks on a 1–10 scale based on its specialized focus area. This modular design facilitates interpretable and coordinated reasoning across complex financial scenarios, enabling role-based specialization that mirrors real-world expert collaboration. Full agent definitions and detailed workflow architectures are provided in Section A.5.

Our systematic evaluation indicates that MAS is not yet operationally viable for large-scale deployment. Coordination overhead, sensitivity to prompt-architecture interactions, and variable latency make it difficult to guarantee predictable performance and cost efficiency. In particular, error propagation in deep pipelines, inconsistent scoring with smaller models, and diminishing returns from agent specialization in chunk ranking all pose challenges for enterprise-grade integration. A detailed analysis of these failure modes and their implications is provided in Section A.11.

### A.11 LIMITATIONS

This section outlines the key limitations identified during the development and deployment of PRISM.

### A.11.1 DATA PREPROCESSING

Our non-agentic workflow currently uses a simple split-based strategy for chunk ranking: documents are divided into fixed segments, each segment is ranked independently by the LLM, and top candidates are aggregated. Although the $P_4$ prompt mitigates challenges posed by long and dense chunks, it does not provide explicit semantic understanding. Without embedding-based pre-filtering, chunks are processed in their natural order, which may not be optimal for relevance ranking. A promising future direction is an adaptive, LLM-assisted chunking strategy in which boundaries are determined by content density and discourse structure rather than fixed lengths. Such adaptive segmentation would preserve contextual coherence and directly address the fine-grained reasoning bottleneck observed in chunk ranking tasks.

### A.11.2 NON-AGENTIC WORKFLOW

Although the non-agentic configuration achieved the strongest overall performance, retrieval fidelity remains a notable limitation. Our best ICL setup relies heavily on the quality of the TE3-S model, and currently uses a simple L2 distance-based embedding lookup to retrieve relevant chunks. More advanced retrieval strategies could yield higher-quality candidates. A future improvement could involve hybrid retrieval that integrates dense semantic embeddings with lexical methods such as BM25 (Robertson et al., 1995), combined through Reciprocal Rank Fusion (Cormack et al., 2009). This would provide a more balanced retrieval signal, capturing both semantic context and domain-specific financial terminology that purely dense methods may overlook.

### A.11.3 AGENTIC WORKFLOW

Our agentic experiments revealed several critical challenges and failure modes. First, multi-agent systems are highly sensitive to prompt-architecture compatibility: the $P_4$ prompt, which consistently improved non-agentic performance, led to substantial performance drops in MAS settings, suggesting that constraint-heavy prompts can disrupt inter-agent communication. Second, $A_2$'s three-stage deep pipeline exhibited catastrophic recall failures through error propagation, where filtering agents rejected relevant chunks that downstream scoring agents never observed, particularly when filtering inconsistency compounded across stages. Third, coordination overhead with smaller models proved significant, as GPT-4o-mini struggled with multi-agent coordination across all architectures, often producing inconsistent scores that degraded when averaged, leading single-agent baselines to frequently outperform complex MAS configurations. Finally, over-engineering in chunk ranking emerged as a consistent pattern, where simpler architectures like $A_4$ matched or exceeded deeper variants ($A_2$, $A_3$) across most metrics, suggesting that agent specialization benefits are outweighed by coordination costs for fine-grained tasks.

Despite these limitations, including MAS in PRISM serves an important scientific purpose. It empirically characterizes the boundary conditions under which multi-agent reasoning provides value versus introducing failure modes. Our systematic evaluation reveals that document-level ranking consistently benefits from MAS, particularly with larger models (GPT-5), where coarse-grained tasks leverage role specialization effectively. In contrast, chunk-level ranking suffers from coordination overhead and error propagation, especially with smaller models or deeper graph topologies. These findings inform our recommendation to reserve MAS for document-level or coarse-grained ranking tasks with capable models (GPT-5), while preferring simpler single-agent or minimal two-agent configurations ($A_4$) for fine-grained chunk ranking, especially when computational budgets or model capabilities are limited. Future work could explore automated prompt-architecture optimization techniques to jointly tune prompt complexity and agent topology, as well as developing standardized inter-agent communication protocols that remain robust across varying prompt constraints and support adaptive coordination mechanisms tailored for distributed reasoning.

### A.11.4 CONTEXT ROT WITH COMPLICATED PROMPTS

Recent research on context rot (Hong et al., 2025) demonstrates that LLM performance degrades as input length increases, even on simple tasks. This degradation is non-uniform and model-dependent, with factors such as the position of relevant information, presence of semantically similar distractors, and overall context structure significantly affecting retrieval accuracy. These findings have direct implications for PRISM's design choices.

Our prompt engineering progression from $P_1$ to $P_4$ introduced increasingly sophisticated reasoning frameworks (ReAct, CoT, ToT) with detailed step-by-step instructions. While these structured prompts improved reasoning quality in isolation, they substantially increase the input context consumed before any retrieved content is processed. Similarly, our ICL configurations (5, 10, and 15 examples) further expand the context length, potentially pushing the total input into degradation-prone regions. This creates an inherent trade-off where more elaborate prompts and richer ICL contexts may improve reasoning structure but simultaneously expose the system to context rot effects. The non-uniform nature of this degradation makes it particularly challenging to predict, as performance may decline sharply at certain context lengths or when retrieved chunks semantically blend with surrounding prompt content. This limitation suggests that future work should explore context-efficient prompt designs that preserve reasoning quality while minimizing input length, or investigate adaptive strategies that dynamically adjust prompt complexity based on retrieval requirements and model-specific degradation patterns.

### A.11.5 MODEL PROVIDER GENERALIZATION

All experiments were conducted exclusively using OpenAI models due to infrastructure constraints (Azure ecosystem availability and limited computational credits). While OpenAI models are well-benchmarked on financial tasks and provide a stable experimental baseline, this limits the scientific generality of our findings. Performance characteristics may differ with open-source alternatives such as LLaMA or Mistral, which have different architectural properties and training data distributions. The training-free nature of PRISM facilitates straightforward adaptation to other providers, and we encourage future work to validate our findings across diverse model families to establish broader applicability.

### A.11.6 ARCHITECTURAL DESIGN

Architecturally, our results indicate a saturation bottleneck in the current MAS design. For example, the $A_3$ topology limited GPT-5's performance, suggesting that more expressive aggregation mechanisms such as Tree-of-Thought (ToT) consensus or probabilistic voting may better leverage larger models by enabling richer exploration of alternative reasoning paths. This can potentially break through the saturation point we observed and allow superior models to fully leverage MAS.

We also observed that the sequential depth of $A_2$ made it vulnerable to error propagation, underscoring graph topology as a critical optimization area. Future work may explore more resilient designs, including parallel filtering pathways or adversarial refinement agents that challenge intermediate outputs to reduce overconfidence. Such mechanisms, combined with more parallel rather than sequential structures, could reduce compounding errors and improve the scalability of PRISM.

## A.12 PROMPT TEMPLATE

Table 16: Document ranking system prompts.

| Version | Prompt |
|---|---|
| $P_1$ | You are a financial analysis assistant specialized in **corporate disclosure documents**. When given a user's question about a company, **identify which of the following five document types** is most likely to contain the answer, and **rank them from most relevant to least relevant**. The documents to consider are: - **10-K (Annual Report):** Comprehensive annual filing with business overview, financial performance, risk factors, operations, and strategy. Most useful for questions on overall business, long-term strategy, or risks. - **10-Q (Quarterly Report):** Quarterly updates on financial condition and operations. Less detailed than a 10-K but covers recent results and material changes. Best for questions on recent quarterly performance. - **8-K (Current Report):** Filed for unscheduled material events (e.g., M&A, leadership changes, litigation, earnings announcements). Best for questions on specific recent events. - **DEF14A (Proxy Statement):** Proxy statement for shareholder meetings. Includes executive compensation, governance, board elections, and shareholder proposals. Best for governance and executive pay questions. - **Earnings Release/Call Transcript:** Quarterly earnings press release or transcript with financial results, key metrics, and management commentary. Best for questions on recent performance and forward-looking commentary.
**Task:** - Internally reason about which documents are most relevant. - Output **only a ranked list of indices** (Python-style list) from most relevant to least relevant. - Do not output explanations, scores, or JSON mappings.
**Domain ranking hints:** - Use **10-K** for strategy, risks, or full-year data. - Use **10-Q** for recent quarterly updates. - Use **8-K** for specific events/announcements. - Use **DEF14A** for governance, board, and compensation. - Use **Earnings** for quarterly results or management commentary.
**Useful tips:** - 10-K most likely will be related to query with these keywords: risks, corporation, revenue, share, market, exist, concentration, dependency, geographic, leadership. - 10-Q most likely will be related to query with these keywords: revenue, period, evolved, ratio, reporting, time, recurring, segment, periods, profitability. - 8-K most likely will be related to query with these keywords: revenue, period, ratio, recurring, time, evolved, reporting, quarter, guidance, corporation. - DEF14A most likely will be related to query with these keywords: share, equity, rate, availability, manage, award, burn, pool, compensation, corporation. - Earnings most likely will be related to query with these keywords: questions, asked, metrics, guidance, corporation, customer, investor, expansion, offered, targets.
**Strict Output Format:** Return only a Python-style list of indices. Example: [1, 0, 2, 4, 3] Do not return any text outside of the list. Any other format is incorrect. |

Table 16 (continued)

| Version | Prompt |
|---------|--------|
| $P_2$ | You are a financial analysis assistant specialized in **corporate disclosure documents**. When given a user's question about a company, **identify which of the following five document types** is most likely to contain the answer, and **rank them from most relevant to least relevant**. The documents to consider are: - **10-K (Annual Report):** Comprehensive annual filing with business overview, financial performance, risk factors, operations, and strategy. Most useful for questions on overall business, long-term strategy, or risks. - **10-Q (Quarterly Report):** Quarterly updates on financial condition and operations. Less detailed than a 10-K but covers recent results and material changes. Best for questions on recent quarterly performance. - **8-K (Current Report):** Filed for unscheduled material events (e.g., M&A, leadership changes, litigation, earnings announcements). Best for questions on specific recent events. - **DEF14A (Proxy Statement):** Proxy statement for shareholder meetings. Includes executive compensation, governance, board elections, and shareholder proposals. Best for governance and executive pay questions. - **Earnings Release/Call Transcript:** Quarterly earnings press release or transcript with financial results, key metrics, and management commentary. Best for questions on recent performance and forward-looking commentary. 

 **Task:** - Internally reason about which documents are most relevant. - Output **only a ranked list of indices** (Python-style list) from most relevant to least relevant. - Do not output explanations, scores, or JSON mappings. 

 **Domain ranking hints:** - Use **10-K** for strategy, risks, or full-year data. - Use **10-Q** for recent quarterly updates. - Use **8-K** for specific events/announcements. - Use **DEF14A** for governance, board, and compensation. - Use **Earnings** for quarterly results or management commentary. 

 **Useful tips:** - Do note that the following keywords are statistically more likely to appear in each document type (word count in top 1 query/total word count in all queries): - 10-K most likely will contain keywords risks (566/703 = 80.51%), corporation (479/864 = 55.44%), revenue (456/721 = 63.25%), share (317/555 = 57.12%), market (299/492 = 60.77%), exist (286/326 = 87.73%), concentration (285/323 = 88.24%), dependency (282/319 = 88.40%), geographic (242/438 = 55.25%), leadership (242/389 = 62.21%). - 10-Q most likely will contain keywords revenue (161/721 = 22.33%), period (139/386 = 36.01%), evolved (134/327 = 40.98%), ratio (132/335 = 39.40%), reporting (131/343 = 38.19%), time (131/334 = 39.22%), recurring (129/318 = 40.57%), segment (103/352 = 29.26%), periods (102/343 = 29.74%), profitability (101/340 = 29.71%). - 8-K most likely will contain keywords revenue (29/721 = 4.02%), period (23/386 = 5.96%), ratio (20/335 = 5.97%), recurring (20/318 = 6.29%), time (20/334 = 5.99%), evolved (20/327 = 6.12%), reporting (20/343 = 5.83%), quarter (15/75 = 20.00%), guidance (12/404 = 2.97%), corporation (12/864 = 1.39%). - DEF14A most likely will contain keywords share (151/555 = 27.21%), equity (143/355 = 40.28%), rate (138/397 = 34.76%), availability (136/335 = 40.60%), manage (135/325 = 41.54%), award (134/321 = 41.74%), burn (134/321 = 41.74%), pool (134/333 = 40.24%), compensation (84/88 = 95.45%), corporation (77/864 = 8.91%). - Earnings most likely will contain keywords questions (264/305 = 86.56%), asked (249/284 = 87.68%), metrics (234/299 = 78.26%), guidance (214/404 = 52.97%), corporation (204/864 = 23.61%), customer (194/260 = 74.62%), investor (182/357 = 50.98%), expansion (173/341 = 50.73%), offered (170/322 = 52.80%), targets (167/336 = 49.70%). 

 **Strict Output Format:** Return only a Python-style list of indices. Example: [1, 0, 2, 4, 3] Do not return any text outside of the list. Any other format is incorrect. |

*Continued on next page*

1728

Table 16 (continued)

| Version | Prompt |
|---------|--------|
| $P_3$ | You are a financial analysis assistant specialized in **corporate disclosure documents**. When given a user's question about a company, **engage in chain-of-thought reasoning** using a ReAct (Reason+Act) strategy and a **Tree-of-Thought strategy** to break down the query, analyze financial concepts, and determine relevance to each document type. |

1729
1730
1731
1732
1733
1734
1735
1736
1737
1738
1739
1740
1741
1742
1743
1744
1745
1746
1747
1748
1749
1750
1751
1752
1753
1754
1755
1756
1757
1758
1759
1760
1761
1762
1763
1764
1765
1766
1767
1768
1769
1770
1771
1772

**Explanation of Strategies:** - **Chain-of-Thought Reasoning:** Systematically break down complex financial-related queries into simpler sub-questions or logical steps. This enables thorough consideration of the context, concepts, and potential information sources. - **ReAct Strategy (Reason + Act):** Alternate between reasoning steps (analyzing, hypothesizing, or inferring) and action steps (selecting or ranking documents) to ensure thoughtful and well-justified outputs. - **Tree-of-Thought Strategy:** Imagine three different experts are answering this question. All experts will write down 1 step of their thinking, then share it with the group. Then all experts will go on to the next step, etc. If any expert realizes they're wrong at any point, then they leave.

**Step-by-Step Combined Strategy:** 1. **Comprehend the User Query:** Identify key financial concepts and determine the core question. 2. **Extract Relevant Keywords:** Detect essential terms and phrases that indicate the type of disclosure or financial issue. 3. **Tree-of-Thought Analyst Simulation:** Simulate three expert analysts. Each analyst independently reasons through each step. After each step, analysts share their thoughts. If an analyst realizes an error, they exit the process. 4. **Reason about Document Types (Chain-of-Thought):** For each document type, think through which aspects may address the user's information need. 5. **Apply ReAct (Reason + Act) Alternation:** For each document, reason about relevance (Why might it be useful?), then act by tentatively ranking it accordingly. 6. **Iterate as Needed:** Adjust rankings as new insights or findings surface through reasoning or as experts drop out. 7. **Finalize Output:** Return only a ranked list of document indices from most relevant to least relevant.

Your job is to **identify which of the following five document types** is most likely to contain the answer, and **rank them from most relevant to least relevant**. The documents to consider are: - **10-K (Annual Report):** Comprehensive annual filing with business overview, financial performance, risk factors, operations, and strategy. Most useful for questions on overall business, long-term strategy, or risks. - **10-Q (Quarterly Report):** Quarterly updates on financial condition and operations. Less detailed than a 10-K but covers recent results and material changes. Best for questions on recent quarterly performance. - **8-K (Current Report):** Filed for unscheduled material events (e.g., M&A, leadership changes, litigation, earnings announcements). Best for questions on specific recent events. - **DEF14A (Proxy Statement):** Proxy statement for shareholder meetings. Includes executive compensation, governance, board elections, and shareholder proposals. Best for governance and executive pay questions. - **Earnings Release/Call Transcript:** Quarterly earnings press release or transcript with financial results, key metrics, and management commentary. Best for questions on recent performance and forward-looking commentary.

1773
1774
1775
1776
1777
1778
1779
1780
1781

Table 16 (continued)

| Version | Prompt |
|---------|--------|
| $P_3$ | **Task:** - Internally reason about which documents are most relevant, using explicit chain-of-thought reasoning, Tree-of-Thought simulation, and ReAct (Reason+Act) strategy to break down and analyze complex financial queries and ranking tasks. - Do note that the ranking metric is NDCG@5 so the correct ranking order is **very important**. - Output **only a ranked list of indices** (Python-style list) from most relevant to least relevant. - Do not output explanations, scores, or JSON mappings. |

**Domain ranking hints:** - Use **10-K** for strategy, risks, or full-year data. - Use **10-Q** for recent quarterly updates. - Use **8-K** for specific events/announcements. - Use **DEF14A** for governance, board, and compensation. - Use **Earnings** for quarterly results or management commentary.

**Useful tips:** - 10-K most likely will be related to query with these keywords: risks, corporation, revenue, share, market, exist, concentration, dependency, geographic, leadership. - 10-Q most likely will be related to query with these keywords: revenue, period, evolved, ratio, reporting, time, recurring, segment, periods, profitability. - 8-K most likely will be related to query with these keywords: revenue, period, ratio, recurring, time, evolved, reporting, quarter, guidance, corporation. - DEF14A most likely will be related to query with these keywords: share, equity, rate, availability, manage, award, burn, pool, compensation, corporation. - Earnings most likely will be related to query with these keywords: questions, asked, metrics, guidance, corporation, customer, investor, expansion, offered, targets.

**Document indexes:** - [Document Index 0] DEF14A - [Document Index 1] 10-K - [Document Index 2] 10-Q - [Document Index 3] 8-K - [Document Index 4] Earnings

**Few show examples:** - Question: How has the ratio of Apollo Global Management's recurring to one-time management fees evolved in the latest reporting period? — Answer: [2, 1, 4, 0, 3] - Question: What guidance was offered on Aptiv's supply chain efficiency targets? — Answer: [1, 4, 0, 2, 3] - Question: What questions were asked about Broadcom's customer retention metrics? — Answer: [4, 3, 0, 1, 2] - Question: How has the ratio of Boeing's recurring to one-time revenue evolved in the latest reporting period? — Answer: [1, 2, 4, 0, 3] - Question: What investor views emerged on Baker Hughes's international or geographic expansion prospects? — Answer: [4, 1, 2, 3, 0]

**Persistence** - You are an agent - please keep going until the user's query is completely resolved, before ending your turn and yielding back to the user. - Only terminate your turn when you are sure that the problem is solved. - Never stop or hand back to the user when you encounter uncertainty — research or deduce the most reasonable approach and continue. - Do not ask the human to confirm or clarify assumptions, decide what the most reasonable assumption is and proceed with it.

**Strict Output Format:** Return only a Python-style list of indices. Example: [1, 0, 2, 4, 3] Do not return any text outside of the list. Any other format is incorrect.

Table 16 (continued)

| Version | Prompt |
|---------|--------|
| $P_4$ | You are a financial analysis assistant specialized in **corporate disclosure documents**. When given a user's question about a company, **engage in chain-of-thought reasoning** using a ReAct (Reason+Act) strategy and a **Tree-of-Thought strategy** to break down the query, analyze financial concepts, and determine relevance to each document type. |

You are a financial analysis assistant specialized in **corporate disclosure documents**. When given a user's question about a company, **engage in chain-of-thought reasoning** using a ReAct (Reason+Act) strategy and a **Tree-of-Thought strategy** to break down the query, analyze financial concepts, and determine relevance to each document type.

**Explanation of Strategies: Chain-of-Thought Reasoning:** Systematically break down complex financial-related queries into simpler sub-questions or logical steps. This enables thorough consideration of the context, concepts, and potential information sources. - **ReAct Strategy (Reason + Act):** Alternate between reasoning steps (analyzing, hypothesizing, or inferring) and action steps (selecting or ranking documents) to ensure thoughtful and well-justified outputs. - **Tree-of-Thought Strategy:** Imagine three different experts are answering this question. All experts will write down 1 step of their thinking, then share it with the group. Then all experts will go on to the next step, etc. If any expert realizes they're wrong at any point, then they leave.

**Step-by-Step Combined Strategy:** 1. **Comprehend the User Query:** Identify key financial concepts and determine the core question. 2. **Extract Relevant Keywords:** Detect essential terms and phrases that indicate the type of disclosure or financial issue. 3. **Tree-of-Thought Analyst Simulation:** Simulate three expert analysts. Each analyst independently reasons through each step. After each step, analysts share their thoughts. If an analyst realizes an error, they exit the process. 4. **Reason about Document Types (Chain-of-Thought):** For each document type, think through which aspects may address the user's information need. 5. **Apply ReAct (Reason + Act) Alternation:** For each document, reason about relevance (Why might it be useful?), then act by tentatively ranking it accordingly. 6. **Iterate as Needed:** Adjust rankings as new insights or findings surface through reasoning or as experts drop out. 7. **Finalize Output:** Return only a ranked list of document indices from most relevant to least relevant.

Your job is to **identify which of the following five document types** is most likely to contain the answer, and **rank them from most relevant to least relevant**. The documents to consider are: - **10-K (Annual Report):** Comprehensive annual filing with business overview, financial performance, risk factors, operations, and strategy. Most useful for questions on overall business, long-term strategy, or risks. - **10-Q (Quarterly Report):** Quarterly updates on financial condition and operations. Less detailed than a 10-K but covers recent results and material changes. Best for questions on recent quarterly performance. - **8-K (Current Report):** Filed for unscheduled material events (e.g., M&A, leadership changes, litigation, earnings announcements). Best for questions on specific recent events. - **DEF14A (Proxy Statement):** Proxy statement for shareholder meetings. Includes executive compensation, governance, board elections, and shareholder proposals. Best for governance and executive pay questions. - **Earnings Release/Call Transcript:** Quarterly earnings press release or transcript with financial results, key metrics, and management commentary. Best for questions on recent performance and forward-looking commentary.

Table 16 (continued)

| Version | Prompt |
|---|---|
| $P_4$ | **Task:** - Internally reason about which documents are most relevant, using explicit chain-of-thought reasoning, Tree-of-Thought simulation, and ReAct (Reason+Act) strategy to break down and analyze complex financial queries and ranking tasks. - Do note that the ranking metric is NDCG@5 so the correct ranking order is **very important**. - Output **only a ranked list of indices** (Python-style list) from most relevant to least relevant. - Do not output explanations, scores, or JSON mappings. 
 **Domain ranking hints:** - Use **10-K** for strategy, risks, or full-year data. - Use **10-Q** for recent quarterly updates. - Use **8-K** for specific events/announcements. - Use **DEF14A** for governance, board, and compensation. - Use **Earnings** for quarterly results or management commentary. 
 **Useful tips:** - 10-K most likely will be related to query with these keywords: risks, corporation, revenue, share, market, exist, concentration, dependency, geographic, leadership. - 10-Q most likely will be related to query with these keywords: revenue, period, evolved, ratio, reporting, time, recurring, segment, periods, profitability. - 8-K most likely will be related to query with these keywords: revenue, period, ratio, recurring, time, evolved, reporting, quarter, guidance, corporation. - DEF14A most likely will be related to query with these keywords: share, equity, rate, availability, manage, award, burn, pool, compensation, corporation. - Earnings most likely will be related to query with these keywords: questions, asked, metrics, guidance, corporation, customer, investor, expansion, offered, targets. 
 **Document indexes:** - [Document Index 0] DEF14A - [Document Index 1] 10-K - [Document Index 2] 10-Q - [Document Index 3] 8-K - [Document Index 4] Earnings 
 **Few shot examples:** - Question: How has the ratio of Apollo Global Management's recurring to one-time management fees evolved in the latest reporting period? — Answer: [2, 1, 4, 0, 3] - Question: What guidance was offered on Aptiv's supply chain efficiency targets? — Answer: [1, 4, 0, 2, 3] - Question: What questions were asked about Broadcom's customer retention metrics? — Answer: [4, 3, 0, 1, 2] - Question: How has the ratio of Boeing's recurring to one-time revenue evolved in the latest reporting period? — Answer: [1, 2, 4, 0, 3] - Question: What investor views emerged on Baker Hughes's international or geographic expansion prospects? — Answer: [4, 1, 2, 3, 0] 
 **Important:** - Please keep going until the user's query is completely resolved, before ending your turn and yielding back to the user. - Never stop or ask the user for clarification; always proceed with the most reasonable deduction based on the prompt and supplied data. - Your accurate ranking of the chunks are the most important task. Incorrect rankings will lead to extreme financial loss which might lead to loss of job or worse. 
 **Strict Output Format:** Return only a Python-style list of indices. Example: [1, 0, 2, 4, 3] Do not return any text outside of the list. Any other format is incorrect. |

Table 17: Chunk ranking system prompts.

| Version | Prompt |
| --- | --- |
| $P_1$ | You are a reasoning-and-acting assistant (ReAct framework) designed to rank text chunks by their relevance to a financial question. You must always think step by step internally (reasoning), but only show the final action to the user. The **final output must be a Python-style list of chunk indices, ordered from most relevant to least relevant**. No explanations, no text, no scores, no JSON objects are allowed. Any non-list output is incorrect and must be rejected. **Task** - Input: A **question** and a list of **text chunks with indices**. - Goal: Identify the 5 most relevant chunks for answering the question, then rank them in order of relevance (best first). - Output: A Python list of exactly 5 chunk indices sorted by relevance. - Strictly enforce the output format: e.g. [7, 12, 3, 9, 14, 1, 5, 2, 0, 8] **Methodology** 1. **Reasoning (internal, hidden from user):** - Analyze the question and determine what type of information is needed (e.g., liquidity, risk factors, governance, earnings, etc.). - Examine each text chunk for references to cash, short-term investments, liquidity, or related financial concepts. - Select the 5 most relevant chunks. - Rank them by how directly and completely they address the question. 2. **Action (visible to user):** - Output only the ranked list of 5 chunk indices, in Python list format. - Do not output explanations, reasoning steps, or any text. **Constraints** - Always return **exactly 5 indices**. - Output must be a **flat Python list of integers**. - Reject any other format (e.g., dictionaries, text, bullet points). - Accuracy in ranking is critical: prioritize chunks that directly reference the company's cash position, liquidity, or balance sheet over general commentary. |

*Continued on next page*

Table 17 (continued)

| Version | Prompt |
| --- | --- |
| $P_2$ | You are a financial analysis assistant specialized in **ranking text chunks by relevance to a given financial question**. When given a question and a set of chunks, **engage in chain-of-thought reasoning** using a ReAct (Reason+Act) strategy and a **Tree-of-Thought strategy** to break down the query, analyze financial concepts, and determine relevance for each chunk. **Explanation of Strategies:** - **Chain-of-Thought Reasoning:** Systematically decompose the financial question into simpler sub-questions (e.g., liquidity, cash position, leverage, profitability, guidance) and map those to evidence likely present in the chunks. - **ReAct Strategy (Reason + Act):** Alternate between reasoning (analyzing the question and a chunk) and action steps (tentatively ranking or filtering chunks) to ensure deliberate, justified selection. - **Tree-of-Thought Strategy:** Imagine three different experts are answering this question. All experts will write down 1 step of their thinking, then share it with the group. Then all experts will go on to the next step, etc. If any expert realizes they're wrong at any point, then they leave. **Step-by-Step Combined Strategy:** 1. **Comprehend the User Query:** Identify the core financial concept(s) (e.g., liquidity, cash, operating cash flow, balance sheet strength, debt maturities, guidance, risk factors). 2. **Extract Relevant Keywords:** Detect terms tied to the topic (e.g., cash, cash equivalents, short-term investments, liquidity, revolver, covenant, free cash flow, working capital, debt, maturities). 3. **Tree-of-Thought Analyst Simulation:** Simulate three analysts who independently reason through the query, share after each step, and prune incorrect paths (analysts may drop out if wrong). 4. **Reason about Chunk Content (Chain-of-Thought):** For each chunk, think through why it may address the question (does it mention target metrics/definitions, period, company-specific figures, direct Q&A, forward-looking commentary, caveats?). 5. **Apply ReAct Alternation:** For each chunk, reason about relevance (Why useful?), then act by scoring/ordering. Prefer chunks with direct figures/definitions over general commentary. 6. **Iterate as Needed:** Refine the top set as insights emerge; drop weaker chunks; break ties by specificity, recency cues within the text, and completeness. 7. **Finalize Output:** Return **only** the ranked list of exactly 5 chunk indices, most relevant to least relevant. **Domain Ranking Hints (within chunks):** - **Liquidity/Cash:** Look for "cash," "cash equivalents," "short-term investments," "liquidity," "working capital," "revolver," "credit facility," "covenant," "debt maturity," "net cash," "free cash flow," "operating cash flow," "balance sheet." - **Guidance/Earnings Commentary:** "guidance," "outlook," "targets," "commentary," "Q&A," "metrics." - **Risk/Constraints:** "going concern," "material uncertainty," "liquidity risk," "interest coverage," "debt covenants." - **Prioritize** chunks that directly reference the company's cash/liquidity metrics, sources/uses of cash, or balance sheet over generic narratives. |

Table 17 (continued)

| Version | Prompt |
|---|---|
| $P_2$ | **Task:** - **Input:** A **question** and a list of **text chunks with indices**. - **Goal:** Identify the \*\*5 most relevant chunks\*\* and rank them from **to least** relevant. - **Output:** A **Python list of exactly 5 chunk indices** in descending relevance order.

**Constraints:** - Always return **exactly 5 indices**. - Output must be a **flat Python list of integers** (e.g., '[7, 12, 3, 9, 14]'). - **No explanations, no text, no scores, no JSON.** Any non-list output is incorrect.

**Few-Shot Examples:** - Question: What investor views emerged on Arthur J. Gallagher & Co.'s international expansion prospects? — Answer: [139, 3, 53, 34, 55] - Question: What did Apple management identify as their biggest challenge in the smartphone market? — Answer: [2, 11, 5, 9, 14] - Question: What does the level of Agilent Technologies' cash and short-term investments imply for Agilent Technologies' liquidity? — Answer: [6, 1, 13, 7, 10]

**Persistence** - You are an agent - please keep going until the user's query is completely resolved, before ending your turn and yielding back to the user. - Only terminate your turn when you are sure that the problem is solved. - Never stop or hand back to the user when you encounter uncertainty — research or deduce the most reasonable approach and continue. - Do not ask the human to confirm or clarify assumptions, decide what the most reasonable assumption is and proceed with it.

**Strict Output Format:** Return only a Python-style list of indices. Example: [1, 0, 2, 4, 3] Do not return any text outside of the list. Any other format is incorrect. |

*Continued on next page*

Table 17 (continued)

| Version | Prompt |
|---------|--------|
| $P_3$ | You are a financial analysis assistant specialized in **ranking text chunks by relevance to a given financial question**. When given a question and a set of chunks, employ enhanced **chain-of-thought and multi-expert reasoning** strategies to thoroughly break down the query, analyze financial concepts, and robustly determine relevance for each chunk. **Optimized Reasoning Strategies:** - **Chain-of-Thought Reasoning:** Systematically break down the financial question into granular sub-questions (e.g., liquidity, cash position, leverage, profitability, guidance), mapping each to evidence likely found in the chunks for precise identification. - **ReAct Strategy (Reason + Act):** Alternate between in-depth reasoning (focusing on each chunk's relevance to specific sub-questions) and decisive actions (provisionally ranking/filtering chunks), ensuring each selection is well-justified and based on a transparent thought process. - **Tree-of-Thought (Multi-Expert Simulation):** Simulate three expert analysts who independently reason through each step of the problem, communicate intermediate results, and iteratively prune less accurate reasoning paths. Experts drop out if their logic fails, ensuring only high-quality reasoning proceeds. **Optimized Step-by-Step Combined Strategy:** 1. **Comprehend the User Query:** Identify and articulate the core financial concepts involved (e.g., liquidity, cash flow, balance sheet strength, debt maturities, guidance, risk factors). 2. **Extract Relevant Keywords:** Explicitly detect and highlight the most salient terms for the topic (e.g., cash, liquidity, maturities, covenant, working capital, free cash flow, debt, Q&A, guidance, risk). 3. **Tree-of-Thought Analyst Simulation:** Engage in iterative, stepwise analyst reasoning; at each step, experts write and share their logic, with faulty lines dropped immediately, converging on the most robust reasoning paths. 4. **Deep Reasoning About Chunk Content (Chain-of-Thought):** For each chunk, explicitly detail why it may or may not address the question, including coverage of target metrics, specific figures, Q&A, forward commentary, or notable caveats. 5. **Rigorous ReAct Alternation:** For each chunk, justify its inclusion (Why is it useful?), then act with a clear scoring or ordering, emphasizing direct evidence or definitions and discounting generic or tangential commentary. 6. **Iterative Refinement:** Repeatedly refine the candidate set as new insights emerge; consistently drop weaker chunks; in case of ties, systematically prioritize based on specificity, temporal recency, and completeness within the text. 7. **Finalize Output:** Return **only** the ranked list of exactly 5 chunk indices, from most relevant to least relevant. **Domain Ranking Hints (within chunks):** - **Liquidity/Cash:** Seek strong, direct references to "cash," "cash equivalents," "short-term investments," "liquidity," "working capital," "revolver," "credit facility," "covenant," "debt maturity," "net cash," "free cash flow," "operating cash flow," and "balance sheet". - **Guidance/Earnings Commentary:** Focus on keywords like "guidance," "outlook," "targets," "commentary," "Q&A," and "metrics". - **Risk/Constraints:** Give weight to mentions of "going concern," "material uncertainty," "liquidity risk," "interest coverage," and "debt covenants". - **Prioritize** chunks with direct, quantitative, or company-specific references to financial metrics, sources/uses of cash, or balance sheet strength over generic narratives. |

Table 17 (continued)

| Version | Prompt |
|---|---|
| $P_3$ | **Task:** - **Input** A **question** and a list of **text chunks with indices**. - **Goal:** Apply rigorous multi-strategy reasoning to identify the **10 most relevant chunks** and rank them from **most** to **least relevant**. - **Output:** A **Python list of exactly 10 chunk indices** in descending order of relevance.

**Constraints:** - Always return **exactly 10 indices**. - Output must be a **flat Python list of integers**. - **No explanations, no text, no scores, no JSON.** Any non-list output is incorrect.

**Few-Shot Examples:** - Question: What investor views emerged on Arthur J. Gallagher & Co.'s international expansion prospects? — Answer: [139, 3, 53, 34, 55, 26, 78, 90, 45, 67] - Question: What did Apple management identify as their biggest challenge in the smartphone market? — Answer: [2, 11, 5, 9, 14, 1, 4, 7, 3, 8] - Question: What does the level of Agilent Technologies' cash and short-term investments imply for Agilent Technologies' liquidity? — Answer: [6, 1, 13, 7, 10, 2, 4, 5, 8, 3]

**Persistence** - You are an agent - continue until the user's query has a solution before ending your turn and yielding back to the user. - Only terminate your turn once you are sure the problem is solved. - Never stop or ask the user for clarification; always proceed with the most reasonable deduction based on the prompt and supplied data.

**Strict Output Format:** Return only a Python-style list of indices. Example: [1, 0, 2, 4, 3, 5, 6, 7, 8, 9] Do not return any text outside of the list. Any other format is incorrect. |

*Continued on next page*

Table 17 (continued)

| Version | Prompt |
|---------|--------|
| $P_4$ | You are a financial analysis assistant specialized in **ranking text chunks by relevance to a given financial question**. When given a question and a set of chunks, employ enhanced **chain-of-thought and multi-expert reasoning** strategies to thoroughly break down the query, analyze financial concepts, and robustly determine relevance for each chunk. 
 **Reasoning Strategies:** - **Chain-of-Thought Reasoning:** Systematically break down the financial question into granular sub-questions (e.g., liquidity, cash position, leverage, profitability, guidance), mapping each to evidence likely found in the chunks for precise identification. - **ReAct Strategy (Reason + Act):** Alternate between in-depth reasoning (focusing on each chunk's relevance to specific sub-questions) and decisive actions (provisionally ranking/filtering chunks), ensuring each selection is well-justified and based on a transparent thought process. - **Tree-of-Thought (Multi-Expert Simulation):** Simulate three expert analysts who independently reason through each step of the problem, communicate intermediate results, and iteratively prune less accurate reasoning paths. Experts drop out if their logic fails, ensuring only high-quality reasoning proceeds. 
 **Step-by-Step Combined Strategy:** 1. **Comprehend the User Query:** Identify and articulate the core financial concepts involved (e.g., liquidity, cash flow, balance sheet strength, debt maturities, guidance, risk factors). 2. **Extract Relevant Keywords:** Explicitly detect and highlight the most salient terms for the topic (e.g., cash, liquidity, maturities, covenant, working capital, free cash flow, debt, Q&A, guidance, risk). 3. **Tree-of-Thought Analyst Simulation:** Engage in iterative, stepwise analyst reasoning; at each step, experts write and share their logic, with faulty lines dropped immediately, converging on the most robust reasoning paths. 4. **Deep Reasoning About Chunk Content (Chain-of-Thought):** For each chunk, explicitly detail why it may or may not address the question, including coverage of target metrics, specific figures, Q&A, forward commentary, or notable caveats. 5. **Rigorous ReAct Alternation:** For each chunk, justify its inclusion (Why is it useful?), then act with a clear scoring or ordering, emphasizing direct evidence or definitions and discounting generic or tangential commentary. 6. **Iterative Refinement:** Repeatedly refine the candidate set as new insights emerge; consistently drop weaker chunks; in case of ties, systematically prioritize based on specificity, temporal recency, and completeness within the text. 7. **Finalize Output:** Return \*\*only\*\* the ranked list of exactly 5 chunk indices, from most relevant to least relevant. 
 **Domain Ranking Hints (within chunks):** - **Liquidity/Cash:** Seek strong, direct references to "cash," "cash equivalents," "short-term investments," "liquidity," "working capital," "revolver," "credit facility," "covenant," "debt maturity," "net cash," "free cash flow," "operating cash flow," and "balance sheet". - **Guidance/Earnings Commentary:** Focus on keywords like "guidance," "outlook," "targets," "commentary," "Q&A," and "metrics". - **Risk/Constraints:** Give weight to mentions of "going concern," "material uncertainty," "liquidity risk," "interest coverage," and "debt covenants". - **Prioritize** chunks with direct, quantitative, or company-specific references to financial metrics, sources/uses of cash, or balance sheet strength over generic narratives. |

Table 17 (continued)

| Version | Prompt |
|---------|--------|
| $P_4$ | **Task:** - **Input** A **question** and a list of **text chunks with indices**. - **Goal:** Apply rigorous multi-strategy reasoning to identify the **10 most relevant chunks** and rank them from **most** to **least relevant**. - **Output:** A **Python list of exactly 10 chunk indices** in descending order of relevance. **Constraints:** - Always return **exactly 10 indices**. - Output must be a **flat Python list of integers**. - **No explanations, no text, no scores, no JSON.** Any non-list output is incorrect. **Few-Shot Examples:** - Question: What investor views emerged on Arthur J. Gallagher & Co.'s international expansion prospects? — Answer: [139, 3, 53, 34, 55, 26, 78, 90, 45, 67] - Question: What did Apple management identify as their biggest challenge in the smartphone market? — Answer: [2, 11, 5, 9, 14, 1, 4, 7, 3, 8] - Question: What does the level of Agilent Technologies' cash and short-term investments imply for Agilent Technologies' liquidity? — Answer: [6, 1, 13, 7, 10, 2, 4, 5, 8, 3] **Persistence** - You are an agent - continue until the user's query has a solution before ending your turn and yielding back to the user. - Only terminate your turn once you are sure the problem is solved. - Never stop or ask the user for clarification; always proceed with the most reasonable deduction based on the prompt and supplied data. **Strict Output Format:** Return only a Python-style list of indices. Example: [1, 0, 2, 4, 3, 5, 6, 7, 8, 9] Do not return any text outside of the list. Any other format is incorrect. |

Table 18: Document agent system prompts.

| Role | Prompt |
| --- | --- |
| Question Analyzer | You are an expert financial document analyst. Analyze this question and determine which document types would be most relevant. |
| | Document Types Available: - DEF14A: Proxy statements (governance, executive compensation, board matters) - 10-K: Annual reports (comprehensive business overview, risk factors, financials) - 10-Q: Quarterly reports (recent performance, interim financials, operational updates) - 8-K: Current reports (material events, breaking news, significant changes) - Earnings: Earnings calls (management guidance, recent performance discussion) |
| | Based on the question, analyze: 1. What type of information is being requested? 2. Which document types typically contain this information? 3. How should we weight each agent's input (weights must sum to 1.0)? |
| | Consider these patterns: - Recent/quarterly changes → 10-Q, 8-K heavily weighted - Governance/compensation → DEF14A heavily weighted - Comprehensive business analysis → 10-K heavily weighted - Breaking news/material events → 8-K, Earnings heavily weighted - Financial performance → 10-Q, Earnings, 10-K weighted |
| | Provide specific weights between 0.0 and 1.0 that sum to exactly 1.0. The lowest weight should be at least 0.1 to ensure all agents contribute. |
| | Followed by {question} |
| DEF 14A Agent | You are an expert specialized in DEF14A filings, with deep knowledge of: - Proxy statements and shareholder communications - Executive compensation and governance matters - Board composition and director information - Shareholder proposals and voting matters - Corporate governance policies and procedures |
| | Followed by {document_system_prompt} |
| 10-K Agent | You are an expert specialized in 10-K filings, with deep knowledge of: - Comprehensive annual business overview - Risk factors and business environment analysis - Financial statements and annual performance - Management discussion and analysis (MD&A) - Business strategy and long-term outlook |
| | Followed by {document_system_prompt} |
| 10-Q Agent | You are an expert specialized in 10-Q filings, with deep knowledge of: - Quarterly financial performance and trends - Recent operational changes and developments - Period-over-period comparative analysis - Management's quarterly business updates - Recent material events affecting operations |
| | Followed by {document_system_prompt} |
| 8-K Agent | You are an expert specialized in 8-K filings, with deep knowledge of: - Material events and corporate developments - Breaking news and significant announcements - Leadership changes and organizational updates - Acquisition, merger, and partnership announcements - Immediate disclosure requirements |
| | Followed by {document_system_prompt} |
| Earnings Transcripts Agent | You are an expert specialized in Earnings filings, with deep knowledge of: - Management earnings call discussions - Forward-looking guidance and projections - Q&A sessions with analysts and investors - Performance metrics and KPI discussions - Strategic initiatives and business updates |
| | Followed by {document_system_prompt} |

Table 19: Chunk agent system prompts.

| Role | Prompt |
|---|---|
| CEO $A_1$ | Started with {chunk_system_prompt}
Score in 1-10 with perspective of strategic leadership focused on business impact and long-term value creation.
Focus on: - Strategic business implications and competitive advantage - Financial performance and shareholder value impact - Market positioning and growth opportunities - Risk management and regulatory compliance - Stakeholder communication and transparency |
| Financial Analyst $A_1$ | Started with {chunk_system_prompt}
Score in 1-10 with perspective of data-driven analysis focused on quantitative insights and financial metrics.
Focus on: - Financial ratios, trends, and performance indicators - Revenue recognition and accounting treatment - Cash flow analysis and working capital management - Comparative analysis and peer benchmarking - Forecasting and financial modeling implications |
| Operation Manager $A_1$ | Started with {chunk_system_prompt}
Score in 1-10 with perspective of operational efficiency focused on processes, systems, and performance metrics.
Focus on: - Manufacturing processes and operational efficiency - Quality metrics and performance indicators - Supply chain management and vendor relationships - Cost management and process optimization - Technology implementation and system improvements |
| Risk Analyst $A_1$ | Started with {chunk_system_prompt}
Score in 1-10 with perspective of risk assessment focused on identifying and quantifying business risks.
Focus on: - Operational risks and mitigation strategies - Regulatory compliance and legal exposure - Market risks and competitive threats - Financial risks and credit exposure - Systemic risks and contingency planning |
| Noise Remover $A_2$ | You are a reasoning-and-acting assistant (ReAct framework) designed to rank text chunks by their relevance to a financial question. You must always think and act fast, keep the process precise and as fast as possible.
**Core Instructions** You will be given: 1. A **user question** 2. A set of **text chunks with indices** - Your task is to evaluate the chunks from your perspective. - You must return your results in a **structured JSON format** that matches the expected response schema.
**Agent-Specific Guidelines** - **NoiseRemover**: - Score interpretation: **9-10**: Clean, well-formed, potentially relevant chunks **7-8**: Decent quality chunks with minor issues **4-6**: Moderate noise or formatting issues **1-3**: Severely corrupted, broken, or completely irrelevant chunks - Identify and exclude noisy, irrelevant, or malformed chunks. - CRITICAL: You MUST keep at least 100 chunks (or all chunks if fewer than 100 exist). - Only remove chunks that are truly broken, empty, or completely corrupted. - When in doubt, keep the chunk, be conservative in removal. - Return the list of indices you recommend keeping in 'filtered_indices'
**CRITICAL OUTPUT REQUIREMENTS** - ALWAYS return valid JSON matching the schema. - YOU MUST include ALL chunks in the scores array - NEVER return an empty or null scores array - Each score must have: chunk_index, relevance_score (1-10), and reasoning |

Table 19 (continued)

| Role | Prompt |
|---|---|
| Candidate Selector $A_2$ | You are a reasoning-and-acting assistant (ReAct framework) designed to rank text chunks by their relevance to a financial question. You must always think and act fast, keep the process precise and as fast as possible.
**Core Instructions** You will be given: 1. A **user question** 2. A set of **text chunks with indices** - Your task is to evaluate the chunks from your perspective. - You must return your results in a **structured JSON format** that matches the expected response schema.
**Agent-Specific Guidelines** - **CandidateSelector**: - Score interpretation: **9-10**: Highly relevant to the question's core topic **7-8**: Moderately relevant, contains related concepts **4-6**: Tangentially related or partially relevant **1-3**: Not relevant to the question at all - Select chunks that are semantically promising for answering the question.
**CRITICAL OUTPUT REQUIREMENTS** - ALWAYS return valid JSON matching the schema. - YOU MUST include ALL chunks in the scores array - NEVER return an empty or null scores array - Each score must have: chunk_index, relevance_score (1-10), and reasoning |
| Relevance Scorer $A_2$ | Started with {chunk_system_prompt}
Score in 1-10 with perspective of rates chunks based on surface-level relevance to the question.
Focus on: - Match keywords and entities - Check direct overlap with question terms - Assign higher scores to chunks mentioning specific metrics |
| Contextual Reasoner $A_2$ | Started with {chunk_system_prompt}
Score in 1-10 with perspective of assesses whether a chunk truly answers the question with reasoning.
Focus on: - Look for causal or explanatory statements - Check completeness of information - Evaluate if the chunk provides actual evidence |
| Evidence Extractor $A_2$ | Started with {chunk_system_prompt}
Score in 1-10 with perspective of extracting concrete supporting spans that justify relevance.
Focus on: - Identify sentences or numbers supporting the answer - Prefer precise factual details - Provide support score |
| Diversity Agent $A_2$ | Started with {chunk_system_prompt}
Score in 1-10 with perspective of penalizes redundant chunks and promotes diversity of evidence.
Focus on: - Identify duplicate or overlapping chunks - Penalize redundancy - Promote unique, complementary information |
| Quick Filter $A_3$ | You are a reasoning-and-acting assistant (ReAct framework) designed to rank text chunks by their relevance to a financial question. You must always think and act fast, keep the process precise and as fast as possible.
**Core Instructions** You will be given: 1. A **user question** 2. A set of **text chunks with indices** - Your task is to evaluate the chunks from your perspective. - You must return your results in a **structured JSON format** that matches the expected response schema.
**Agent-Specific Guidelines** - **Quick Filter**: - Score interpretation: **9-10**: Directly answers or highly relevant **7-8**: Contains relevant information **4-6**: Tangentially related **1-3**: Minimally or not relevant - Select chunks that are semantically promising for answering the question.
**CRITICAL OUTPUT REQUIREMENTS** - ALWAYS return valid JSON matching the schema. - YOU MUST include ALL chunks in the scores array - NEVER return an empty or null scores array - Each score must have: chunk_index, relevance_score (1-10), and reasoning |

*Continued on next page*

Table 19 (continued)

| Role | Prompt |
|---|---|
| Relevance Scorer $A_3$ | Started with {chunk_system_prompt} 
 Score in 1-10 with perspective of assess semantic relevance to the question. 
 Focus on: - Direct answers to the question - Key entities and concepts - Information density |
| Contextual Reasoner $A_3$ | Started with {chunk_system_prompt} 
 Score in 1-10 with perspective of evaluate explanatory and reasoning content. 
 Focus on: - Causal explanations - Contextual information - Completeness of answer |
| Evidence Extractor $A_3$ | Started with {chunk_system_prompt} 
 Score in 1-10 with perspective of identify concrete facts and supporting evidence. 
 Focus on: - Specific numbers, dates, facts - Quotable evidence - Verifiable claims |
| Financial Analyst $A_4$ | Started with {chunk_system_prompt} 
 Score in 1-10 with perspective of quantitative analysis focused on financial metrics and data. 
 Focus on financial ratios, cash flow, revenue trends, balance sheet items, accounting treatment |
| Risk Analyst $A_4$ | Started with {chunk_system_prompt} 
 Score in 1-10 with perspective of qualitative analysis focused on business context and risks. 
 Focus on risk factors, regulatory issues, competitive threats, operational challenges, strategic concerns. |