# OpenReview forum: "PRISM: Prompt-Refined In-Context System Modeling for Financial Retrieval"
_ICLR.cc/2026/Workshop/AFAA — Submitted to AFAA 2026_

### Official Review · Reviewer_XbLD · 2026-02-11
**A multi-agent system for financial retrieval**

**Rating:** 4
**Confidence:** 4

**Summary:**

This paper proposes PRISM, a training-free multi-agent framework for financial information retrieval. The framework combines three components: refined system prompts, in-context learning (ICL) with exemplar retrieval, and optional multi-agent coordination. I view the main contribution as a systematic empirical study that analyzes when each component helps.

The authors evaluate PRISM on FinAgentBench, FiQA-2018, and FinanceBench. On FinAgentBench, PRISM achieves competitive NDCG@5 performance (0.71818) without fine-tuning. The study shows that well-designed prompts and selectively applied document-level ICL consistently improve ranking. In contrast, multi-agent systems only help when larger models are used and when the scope is limited. Overall, the paper positions itself as a practical guide for designing training-free LLM-based financial retrieval systems.

**Strengths:**

1. The paper provides a systematic and controlled empirical study. The ablation in Table 3 clearly separates the effects of prompt design, ICL scope, and multi-agent architectures. This makes the conclusions easy to interpret. I appreciate the comprehensive analysis and experiments.

2. The finding about selective ICL is practically useful. Applying ICL at the document level helps, while extending it to the chunk level hurts performance due to context overload. This observation is non-trivial and sounds intriguing.

3. The paper reports feasibility metrics such as latency, token usage, and cost. Many retrieval papers focus only on accuracy. Here, the authors provide a clear picture of deployment trade-offs, especially for long financial documents. I personally like this part for it reports important metrics for real-world applications, especially for financial retrieval.

4. Negative results on MAS are reported. The experiments show that deeper agent pipelines suffer from error propagation and coordination overhead, especially with smaller models. This challenges the assumption that more agentic structure always improves retrieval.

**Weaknesses:**

The main limitation is methodological novelty. PRISM combines prompt engineering, ICL, and multi-agent workflows, but it does not introduce anything new, like na ew retrieval algorithm or theoretical insight. The contribution lies mainly in empirical findings.

The MAS analysis, while extensive, remains somewhat descriptive. The paper shows that deeper pipelines suffer from over-filtering and error propagation, but it does not provide a deeper failure analysis. For example, how often do early filters remove relevant chunks? What types of queries benefit from MAS? More qualitative error analysis would strengthen the claim.

Also, portability remains unclear. The authors state that the framework is training-free and adaptable, but no experiments are conducted with open-source LLMs or alternative embeddings. As a result, the conclusions may be specific to the OpenAI / property models rather than universal across open-source models.

---

### Official Review · Reviewer_jmuT · 2026-02-14
**Well-Executed and Comprehensive Empirical Study of a Training-Free Financial IR Framework**

**Rating:** 4
**Confidence:** 5

**Summary:**

The paper presents PRISM, a comprehensive and training-free framework for financial information retrieval composed of three complementary (or selectively applied) modules based on prompt engineering, in-context learning, and a multi-agent setup.

The study investigates how recent GPT models can be enhanced for information retrieval in the financial domain and motivates the construction of the framework through an extensive and well-conducted exploratory data analysis. The proposed approach is evaluated through a thorough ablation study across three widely recognized financial IR datasets, strengthening the validity of the conclusions.

A key contribution of the work lies in its systematic analysis of when each module is beneficial (or not), as well as its examination of feasibility aspects, including latency, token efficiency, and reproducibility. The authors further support their findings with statistical analyses, which increases confidence in the reported results.

Overall, the study provides meaningful insights that contribute to the informed design of financial IR systems, including those incorporating agent-based architectures.

**Strengths:**

- The work is well motivated and grounded in recent literature.
- The framework is evaluated through an extensive ablation study across three reference datasets in financial IR, which strengthens the empirical findings.
- The study conducts an analysis of feasibility and reproducibility, which is particularly important in the context of fairness and the increasing non-determinism of LLM- and agent-based systems. The results collectively suggest that the framework achieves stable and reproducible performance.
- The proposal of a training-free framework achieving strong empirical results is especially valuable given frequent resource constraints.
- It is particularly valuable to observe how traditional exploratory data analysis techniques are employed to inform the enhancement of modern LLM-based systems in a principled way. While the exploratory analysis is performed on a single dataset (FINAGENTBENCH), the derived insights effectively guide experimental decisions across the other benchmarks, indicating a meaningful degree of cross-dataset generalization.

**Weaknesses:**

- The framework is evaluated exclusively on OpenAI GPT models, leaving open questions regarding the generalization of the findings to other models, including open-source alternatives. This represents a promising direction for future work.
- Closely related to the previous point, multi-agent systems are often model-dependent and may rely on the agentic capabilities of specific architectures. As such, some findings, particularly those concerning multi-agent setups, may not generalize across models. The presentation of results could benefit from explicitly contextualizing conclusions as applying ``among the evaluated models'', especially regarding model-size dependencies.
- A more minor limitation concerns methodological novelty. While the system is carefully designed and well supported by prior literature on prompting, in-context learning, and agent-based systems, it does not introduce fundamentally new methodological components. An interesting extension would be to leverage the learned characterizations to design a meta-module that dynamically selects and executes the most appropriate configuration at inference time. That said, the extensive characterization and systematic analysis still constitute an important contribution to the field.

---

### Official Review · Reviewer_T836 · 2026-02-19
**PRISM: Prompt-Refined In-Context System Modeling for Financial Retrieval**

**Rating:** 2
**Confidence:** 4

**Summary:**

This paper presents PRISM - a training free framework for financial information retrieval. It works using prompting, in-context learning and also through a multi agent setup. The authors evaluate and study various prompts, and in-context learning techniques and also evaluate different multi-agent architectures for financial benchmarks. The provide metrics on token usage, latency and costs.

**Strengths:**

The following are the strenghs:
1) Clear comparitive study between different In-context learning approaches and also different architectures and also real world usage metrics so this paper is grounded in practicality.
2) They also show that multi agent architectures don't always outperform simple model calling, showing that complexity does not always equal performance.

**Weaknesses:**

1) Though the paper is practical, its not very novel as there are other papers showcasing the same findings - study between ICL, promoting and different architectures is not very novel.
2) This paper does not fit in the workshop theme - it works more as a IR paper and not for a workshop based on fairness.
3) GPT models are already generally very good at multi turn settings, it would have been useful to see the performance gaines on an open source model perform in this frame work to be accepted as a general framework. Even the limitations might have been different if other models other than GPT were studied,

---

### Meta-Review · Area_Chair_LEob · 2026-02-27

**Recommendation:** Main Papers Track
**Confidence:** 5

**Metareview:**

The submission presents PRISM, a training-free framework that integrates refined system prompting, in-context learning (ICL), and lightweight multi-agent coordination for document and chunk ranking tasks. They investigate how recent GPT models can be enhanced for information retrieval in the financial domain. The authors evaluate various prompts, in-context learning techniques, and different multi-agent architectures for financial benchmarks through exploratory data analysis. The study and findings will be of great use to practitioners.

---

### Decision · Program_Chairs · 2026-03-02

**Decision:**

Reject

**Comment:**

As highlighted by the reviewers, the paper unfortunately does not align with the themes of the workshop. Thus, despite strong contributions, the paper has been rejected from the workshop. We encourage the authors to take advantage of the reviews, and submit their work at a more appropriate venue.